

# MIPAS ozone retrieval version 8: middle atmosphere measurements

Manuel López-Puertas[1], Maya García-Comas[1], Bernd Funke[1], Thomas von Clarmann[2], Norbert Glatthor[2], Udo Grabowski[2], Sylvia Kellmann[2], Michael Kiefer[2], Alexandra Laeng[2], Andrea Linden[2], and Gabriele P. Stiller[2]

[1]Instituto de Astrofísica de Andalucía, CSIC, Granada, Spain
[2]Karlsruhe Institute of Technology, Institute of Meteorology and Climate Research, Karlsruhe, Germany

**Correspondence:** M. López-Puertas (puertas@iaa.es)

**Abstract.** We present a new version of $O_3$ data retrieved from the three MIPAS observations modes of the middle atmosphere (MA, UA and NLC). The $O_3$ profiles cover altitudes from 20 up to 100 km altitudes for daytime and up to 105 km at nighttime, for all latitudes, and the period 2005 until 2012. The data has been obtained with the IMK–IAA MIPAS level 2 data processor and are based on ESA version 8 re-calibrated radiance spectra with improved temporal stability. The processing included

several improvements with respect to the previous version, such as the consistency of the microwindows and spectroscopic data with those used in the nominal mode V8 data, the $O_3$ a priori profiles, and updates of the non-LTE parameters and of the nighttime atomic oxygen. Random errors are dominated by the measurement noise with $1\sigma$ values for single profiles for daytime of <5% below ~60 km, 5–10% between 60 and 70 km, 10–20% at 70–90 km and about 30% at 95 km. For nighttime, they are very similar below 70 km but smaller above (10–20% at 75–95 km, 20–30% at 95–100 km and larger than 30% above

100 km). The systematic error is ~6% below ~60 km (dominated by uncertainties in spectroscopic data), and 8–12% above ~60 km, mainly caused by non-LTE uncertainties. The systematic errors in the 80-100 km range are significantly smaller than in the previous version. The major differences with respect to the previous version are: 1) The new retrievals provide $O_3$ abundances in the 20 – 50 km altitude range larger by about 2–5% (0.2–0.5 ppmv); 2) $O_3$ abundances reduced by ~2–4% between 50 and 60 km in the tropics and mid-latitudes; 3) reduced $O_3$ abundances in the nighttime $O_3$ minimum just below

80 km, leading to a more realistic diurnal variation; 4) larger daytime $O_3$ concentrations in the secondary maximum at the tropical and mid-latitudes (~40%, 0.2–0.3 ppmv); and 5) nighttime $O_3$ abundances in the secondary maximum reduced by 10–30%. The $O_3$ profiles retrieved from the nominal mode (NOM) and the middle atmosphere modes are fully consistent in their common altitude range (20–70 km). Only at 60–70 km daytime $O_3$ of NOM seems to be larger than that of MA/UA by 2–10%. Compared to other satellite instruments, MIPAS seems to have a positive bias of 5-8% below 70 km. Noticeably,

the new version of MIPAS agrees much better than before with all instruments in the upper mesosphere/lower thermosphere, reducing the differences from ~ ±20% to ~ ±10%. Further, the diurnal variation of $O_3$ in the upper mesosphere (near 80 km) has been significantly improved.



# 1   Introduction

Ozone is an essential variable of the atmosphere. In addition to absorbing the harmful solar UV radiation, it also plays key
roles in the energy balance and chemistry of the middle and upper atmosphere (Brasseur and Solomon, 2005). Because of its
importance, it has been extensively measured in the stratosphere and in the mesosphere using different techniques. Kaufmann
et al. (2003) and Smith et al. (2013) have reported very comprehensive reviews of $O_3$ observations; the former focused on those
carried out before 2003, and the latter concentrated on the satellite observations performed more recently. Measurements, that
cover the three typical ozone maxima and, simultaneously, its diurnal variation with global latitudinal coverage are, however,
scarce.

The Michelson Interferometer for Passive Atmospheric Sounding (MIPAS), a high spectral resolution limb sounder on board
the Envisat satellite, measured the atmospheric $O_3$ from March 2002 until April 2012. The MIPAS interferometer had an ample
spectral coverage and a very high spectral resolution. It was operated at $0.025\,cm^{-1}$ during 2002-2004 and at $0.0625\,cm^{-1}$ from
2005 until April 2012, and scanning the atmosphere from pole-to-pole during day (10 am) and night (10 pm) (Fischer et al.,
2008). As its major objective was to understand the stratospheric chemistry and dynamics, it was observing most of the time
the 6-68 km altitude range (the so-called nominal mode or 'NOM'. However, mainly after 2005, it was also regularly pointing
at higher altitudes in its middle atmosphere (MA), noctilucent (NLC), and upper atmosphere (UA) modes (De Laurentis, 2005;
Oelhaf, 2008).

The retrieval of ozone from the NOM mode has been carried out, among others, by the Institute of Meteorology and Climate
Research and Instituto de Astrofísica de Andalucía (IMK/IAA) (Glatthor et al., 2006) and the most recent retrieval version from
L1b V8.03 MIPAS spectra is described by Kiefer et al. (2023). The retrieval of $O_3$ from the MIPAS middle and upper atmo-
sphere modes (including MA, UA and NLC), version V5R_O3_m22[1], has been reported by López-Puertas et al. (2018). The
major difference between the retrievals of the NOM and the middle/upper atmosphere modes is the necessity of incorporating
accurate non-local thermodynamic equilibrium (non-LTE) in the radiative transfer calculations for the latter three.

In 2018, a new version, V8.03, of MIPAS L1b spectra became available. This version supersedes the preceding versions.
In particular, version 5, the latest used in the middle/atmosphere retrievals, which suffered from an inadequate nonlinearity-
correction of the gain calibration, that led to an artificial drift in ozone (Eckert et al., 2014; Kiefer et al., 2023). For that
reason, together with the fact that this version would possibly be the last one for some time, we reprocessed the MIPAS middle
and upper atmosphere $O_3$ data. Thus, the aim of this paper is to document the new ozone data set, V8R_O3_m61, covering
measurements recorded in the MA, UA, and NLC measurement modes. For simplicity, we refer to all three of these data sets
as 'MIPAS middle atmosphere' data. They cover the period from 2005 to 2012 and the altitude ranges of 20–102 km for MA,
40–102 km for UA and 40–102 km for NLC.

The characteristics of the new L1b V8.03 spectra are discussed in Kleinert et al. (2018) and Kiefer et al. (2021). The retrieval
method of the $O_3$ middle atmosphere observations has been described already in Gil-López et al. (2005) and López-Puertas
et al. (2018). In this version, we have also improved and updated the $O_3$ retrieval in several aspects (see Sec. 2). Some of the

---

[1] 'm' is equal to 5, 6 and 7 for the MA, UA and NLC modes, respectively.



updates are common to the $O_3$ retrieved from the nominal mode measurements, obtained up to ~70 km (Kiefer et al., 2023). Others are more specific to the MA, UA, and NLC modes, and are based mainly on the systematic differences found in the validation of V5R_O3_m22 $O_3$ data (López-Puertas et al., 2018). These updates include the $O_3$ a priori data, the revision of the non-LTE processes, and the atomic oxygen concentration during nighttime. We should note that having in mind the possible

future merging of the middle atmosphere and NOM $O_3$ data, whenever possible we have maintained the same retrieval setup as in the $O_3$ NOM dataset A comparison of the new results with the previous version V5R_O3_m22 is presented in Sec. 5, and the consistency between the NOM and MA V8 data is presented in Sec. 6.

The comparison of the middle atmosphere MIPAS $O_3$ data with respect to other satellite data (SABER, GOMOS, MLS, SMILES and ACE-FTS) has been redone, including the more recent available versions of ACE-FTS and MLS (see Sec. 7).

## 65  2  The retrieval of $O_3$ V8R_O3_m61

The MIPAS middle atmosphere V8R_O3_m61 $O_3$ retrieval is based on a constrained nonlinear least squares fitting of limb radiances. It is performed by using the IMK/IAA level 2 Scientific Processor (von Clarmann et al., 2003, 2009) supplemented with the GRANADA (Generic RAdiative traNsfer AnD non-LTE population Algorithm) algorithm (Funke et al., 2012) in order to account for non-LTE emissions. The different aspects of the retrieval, including the basic equations, error estimates,

averaging kernels, the iteration and convergence criteria and the regularization method are described in von Clarmann et al. (2003, 2009) with recent updates in Kiefer et al. (2021, 2023). The details of the retrieval under non-LTE conditions are described by Funke et al. (2001) and the detailed non-LTE model for the $O_3$ vibrational levels in López-Puertas et al. (2018).

As mentioned above, the major motivation of this version of $O_3$ middle atmosphere data is the availability of the new level-1b radiance spectra of ESA, version 8.03, hereafter 'V8', while in the previous version, we used version V5 (5.02/5.06) of the

ESA-calibrated spectra. The major changes applied to ESA V8R L1b data are described in Kiefer et al. (2021, 2023). One of the major improvements is a better calibration introduced by a more adequate correction of the detector's non-linearities in the gain calibration. Time series of $O_3$ retrieved in the previous version V5 were shown to be affected by unrealistically large and negative drifts (Eckert et al., 2014; Hubert et al., 2020). The drifts of the ozone interim version 7, based on level 1 data that already accounted for a more adequate treatment of the gain calibration, were partially reduced but over-corrected (Laeng

et al., 2018). The use of level 1b V8 spectra is expected to ameliorate this.

We recall that the MIPAS spectra used here, taken in the MA, UA, and NLC modes, cover limb tangent heights of 20–102 km at 3 km steps for MA; of 42–172 km at 3 km steps below 102 km and 5 km above, for UA; and of 39–102 km with 3-km steps in the 39–78 km and 87–102 km ranges and 1.5 km at 78–87 km for the NLC mode (De Laurentis, 2005; Oelhaf, 2008). The MIPAS horizontal field of view (FOV) is approximately 30 km. All these measurements were taken at the "reduced" spectral

resolution of 0.0625 cm$^{-1}$ and the vertical field of view is ~3 km.



## 2.1 Retrieval updates common to the nominal and middle atmosphere measurement modes

Many of the updates applied to the ozone retrieval from the nominal-mode measurements (Kiefer et al., 2023) apply also to the middle atmosphere retrievals presented here. For completeness, they are briefly described in this section. Those more specific to the middle atmosphere modes (MA, UA, NLC) are described in Sec. 2.2.

### 2.1.1 Retrieved temperatures

Following the usual sequence of retrieval steps, the temperature and the tangent altitude information retrieved in a previous step from V8R spectra are included in the $O_3$ retrieval. This version of the temperature retrieval includes, in addition to the level-1b spectra, several updates and improvements with respect to the V5R version. Among them, the most important are: the a priori temperature above 43 km (updated from NRLMSISE-00 to SD-WACCM4), the atomic oxygen (updated from the NRLMSISE-00 model to the SD-WACCM4 model at MIPAS geolocations and corrected with MIPAS V5R climatology); and the $CO_2$ concentration now taken from the SD-WACCM4 climatology (see, Kiefer et al., 2021; García-Comas et al., 2023, for more details).

In nominal mode retrievals, where high-altitude temperatures depend largely on a priori assumptions, the new temperature retrieval led to a significant improvement of polar lower mesospheric $O_3$ during large stratospheric warnings (Kiefer et al., 2023). This effect, however, is small in the current MA mode $O_3$, because in this mode the retrieved temperature at higher altitudes does not depend that much on a priori assumptions.

A further improvement of the preceding retrieval of temperature in V8 is the inclusion of its variability along the line-of-sight (LOS). This was considered by including an a priori 3D field with its horizontal structure provided by a priori information and its vertical structure scaled by the retrieved temperature (Kiefer et al., 2021). Following our sequential retrieval approach, the resulting 3D temperature field was used in the forward calculations of $O_3$ limb radiances. Likewise, this temperature field was used to correct the non-LTE populations along the LOS. In the previous $O_3$ version, the variations of temperature along the LOS were implemented more approximately by retrieving a 2–points horizontal temperature gradient around the tangent point (Kiefer et al., 2021).

### 2.1.2 Background continuum, radiance offset and water vapour interference

The IMK/IAA processor retrieves, jointly with the $O_3$ abundance, a background continuum radiation and a zero-level calibration offset. The continuum is retrieved at each microwindow (MW) and altitude; while the offset is assumed to be frequency-independent for the microwindows within each of the A and AB MIPAS bands but retrieved in altitude.

The uppermost altitude of the continuum retrieval has been extended from 50 to 60 km. The background continuum is strongly constrained to zero above 60 km (we used 50 km in the previous version), and the vertical offset profiles for the microwindows in band A (MWs #1–30) and band AB (MWs #31–38) (see Table A1) are strongly regularised towards the a priori values taken from the empirically determined offset correction profiles by Kleinert et al. (2018). See more details in Kiefer et al. (2023).



Further, in this version the abundance of $H_2O$ is also jointly retrieved with those parameters. This is done in order to avoid the propagation of uncertainties of the a priori $H_2O$. Note, however, that the retrieved $H_2O$ profiles are not used further because

they are of sub-optimal quality since the spectral range was selected for the retrieval of ozone but not for $H_2O$ (Kiefer et al., 2023).

### 2.1.3 Microwindows and spectroscopy

As for temperature and other gases, in the retrieval of $O_3$ we use small spectral regions (microwindows) covering ro-vibrational emissions of the main $O_3$ isotope. These MWs vary with tangent altitudes in order to minimise errors and optimise computation

time. Glatthor et al. (2018) has shown that the $O_3$ spectroscopic data in MIPAS band A (685–980 $cm^{-1}$) and those in the AB-band (1010–1180 $cm^{-1}$) are inconsistent. Further, Laeng et al. (2014, 2015) found that the use of microwindows in the AB-band, instead of in band A, leads to a positive bias in MIPAS ozone profiles in the upper troposphere and lower stratosphere. Thus, whenever possible, we used MWs in band A. However, in the mesosphere and lower thermosphere, the $O_3$ lines in band A are too weak and the spectra are noisy; hence the use of MWs in the AB-band was necessary.

The MWs used in this version are essentially the same as in the previous version, and are listed in Table A1 for easy reference. Only minor changes were made. In particular, MW#8 (720.75-723.688 $cm^{-1}$) was excluded due to $CO_2$ line mixing, and MWs #37 and #38 near 1053 and 1055 $cm^{-1}$ were slightly reduced in order to remove the radiance contribution of $CO_2$ laser band lines. The MWs used below 50 km are the same as those used in the NOM $O_3$ retrieval (all located in the A band) and those used between 50 and 70 km are very similar (both from the AB band) (Kiefer et al., 2023).

The $O_3$ spectroscopic data from the HITRAN 2008 database (Rothman et al., 2009) used in the previous version of the $O_3$ retrieval were replaced with those of the MIPAS pf3.2 database (Flaud et al., 2003a,b). This change was motivated because the MIPAS spectroscopic database has smaller inconsistencies between the line strengths in the spectral range of the MIPAS A and AB bands and because in the HITRAN dataset there is an unrealistic large change in the air broadening coefficients near 797 $cm^{-1}$ (Glatthor et al., 2018). For the interfering species, we used HITRAN 2016 (Gordon et al., 2017). $CO_2$ line-mixing

coefficients have been re-calculated for the new spectroscopic data.

### 2.1.4 Other changes in the forward model and inputs parameters

Another less important change included in this version is the concentrations of the interfering species. While in the previous version we used a specifically tailored climatology, here we constructed the interfering species dataset on the basis of previous V5 version results (see Kiefer et al., 2021). The $CO_2$ concentration is taken from the SD-WACCM4 climatologies (see, Kiefer

et al., 2021; García-Comas et al., 2023).

In the current version, we used in the forward model an internal spectroscopic grid of $10^{-3}$ $cm^{-1}$ instead of the slightly finer grid of $5 \times 10^{-4}$ $cm^{-1}$ used in the previous version. We have tested that the effect on the retrieved $O_3$ is negligible while it allows us to reduce the CPU time consumed by the non-LTE retrieval. On the other hand, the following measures increased CPU time: the spectral grid on which the absorption cross sections are calculated has been improved, from 0.00125 $cm^{-1}$ to

0.0009765625 $cm^{-1}$, and the apodization of calculated spectra used a wider frequency range.





## 2.2 Retrieval updates specific to middle atmosphere measurement modes

In this section we describe the updates which are more relevant or exclusive to the retrieval of $O_3$ from the MA, UA and NLC observation modes, and hence are more relevant for the $O_3$ retrieved above $\sim 70$ km.

### 2.2.1 Regularisation

The retrievals are performed from the surface to 120 km over a fixed altitude grid of 1 km up to 50 km, at 72-75 km, and at 77-88 km; of 2 km at 50-72 km, 75-77 km, and 88-102 km; and of 5 km from 105 up to 120 km. Contrary to the retrievals from the nominal observation mode (Kiefer et al., 2023), which are linear in volume mixing ratio (VMR), in the middle atmospheric measurement modes we retrieve the logarithm of VMR. Forward calculations are performed using the same grid. As this grid is finer than the MIPAS vertical sampling of 3 km (except in the NLC mode at 78–87 km where it is 1.5 km), we used a

regularisation; a Tikhonov-type first order smoothing constraint in our case (Tikhonov, 1963). The regularisation used in this version has not changed since version 5.

### 2.2.2 A priori $O_3$

The a priori $O_3$ in the mesosphere and lower thermosphere has changed significantly and affects principally the upper mesospheric daytime $O_3$ mixing ratio. We recall that, with the Tikhonov regularization chosen, the a priori does not affect the total

ozone amount retrieved but only the shape of the vertical profile. In the previous version we used the 2D field from Garcia and Solomon (1994) which had the $O_3$ daytime secondary maximum at altitudes lower than recent measurements (López-Puertas et al., 2018). Here, the $O_3$ a priori is essentially a MIPAS $O_3$ V5 zonal mean climatology (obtained from MA and UA monthly composites, $10°$ resolution, and linearly interpolated in between). This climatology has been modified in order to account for the poorer MIPAS vertical resolution in the upper mesosphere by correcting with $O_{3,\mathrm{corr}} = O_{3,\mathrm{uncorr}} \times [O_{3,\mathrm{W}}/O_{3,\mathrm{W\_AK}}]$,

where $O_{3,\mathrm{W}}$ is taken from the SD-WACCM4 climatology, the Whole Atmosphere Community Climate Model Version 4 simulations (Marsh, 2011; Marsh et al., 2013) of a specified dynamics run (Garcia et al., 2017), and $O_{3,\mathrm{W\_AK}}$ is the same $O_{3,\mathrm{W}}$ climatology but with the MIPAS averaging kernels applied. With this correction, the MIPAS climatology remains essentially unaltered below $\sim 70$ km. Above, the overall magnitude of the MIPAS V5 ozone abundances is maintained, while unresolved profile features are incorporated from the WACCM climatology.

### 2.2.3 Non-LTE: collisional and reaction rates

The ozone non-LTE model used in this version is described in detail by López-Puertas et al. (2018); see Sec. 2 and Tables 1 and 2 in that work. We describe here the major changes in the model that apply to this version. First, we discuss the collisional rates and later on the species abundances required by the non-LTE model and not measured by MIPAS, e.g., the atomic oxygen and hydrogen concentrations.

180       The major uncertainty in the $O_3$ non-LTE model is the deactivation of $O_3(v_1,v_2,v_3)$ by atomic oxygen, either by chemical quenching, $O_3(v_1,v_2,v_3) + O \rightarrow O_2 + O_2$, or by inelastic collisions, $O_3(v_1,v_2,v_3) + O \rightarrow O_3(v_1',v_2',v_3') + O + \Delta E$. The





**Table 1.** Major photochemical reactions affecting $O_3$ in the mesosphere and lower thermosphere.

| No. | Rate | Process |
| --- | --- | --- |
| 1 | $k_1$ | $O_2 + O + M \rightarrow O_3(v_1, v_2, v_3)$ |
| 2 | $k_2$ | $H + O_3 \rightarrow OH^*(v) + O_2$ |
| 3 | $k_3$ | $O + O_3 \rightarrow O_2 + O_2$ |
| 4 | $J_{O3}$ | $O_3 + h\nu \rightarrow O_2 + O$ |

uncertainty comes not only from the rates themselves but also from the uncertain amount of atomic oxygen, which is not measured but derived or constrained by the retrieved $O_3$ in an iterative process (López-Puertas et al., 2018; Mlynczak et al., 2013). For larger rates or larger O concentrations, the deactivation is stronger which leads to lower populations for the emitting $O_3$ states and hence larger $O_3$ abundances.

In the validation of the previous version of $O_3$ we found that the nighttime upper mesospheric $O_3$ was larger by $\sim$20% than in most of the other instruments (López-Puertas et al., 2018). Thus, in order to get a better agreement, we reduced the relaxation of $O_3$ by O, $O_3(v_1, v_3) + O$, by a factor of two, e.g., $4.65 \times 10^{-12}$ cm$^3$s$^{-1}$. Note that the chemical quenching was already neglected in the previous version, as suggested by West et al. (1978). The assumed rate is within the uncertainties of the laboratory measurements but at the lower limit measured by West et al. (1976) (see the discussion in López-Puertas et al., 2018). More recently, Castle et al. (2014) have measured the thermal relaxation of the lower-energy $v_2$ mode[2] in collisions with O at room temperature, finding a value of $(2.2 \pm 0.5) \times 10^{-12}$ cm$^3$s$^{-1}$. Our selected value is still reasonable because, although still larger, it is closer to the relaxation of the lower-energy $v_2$ mode, whose relaxation is expected to be more efficient.

### 2.2.4 Trace gas concentrations relevant for non-LTE modelling

As shown above, the inversion of $O_3$ under non-LTE conditions requires knowledge of the atomic oxygen concentration, [O]. In our retrieval, we constrain [O] by assuming that $O_3$ is in photochemical equilibrium with O, and using the $O_3$ abundance retrieved in the previous iteration of the inversion (daytime) or from the previous version (nighttime). This is a reasonable approach above around 60 km during daytime and $\sim$80 km at nighttime because of the rapid timescales for ozone production and loss in this region. Below those altitudes collisions with O are negligible.

**Daytime**

Taking into account the major photochemical reactions affecting $O_3$ in the mesosphere and lower thermosphere (see Table 1), the O concentration for daytime conditions can be well approximated by

$$[O]_d = \frac{(J_{O3} + k_2[H])\,[O_3]}{k_1\,[O_2]\,[M] - k_3[O_3]} \approx \frac{J_{O3}\,[O_3]}{k_1\,[O_2]\,[M] - k_3[O_3]}, \tag{1}$$

---

[2]The 9.6 $\mu$m emission used here to retrieve $O_3$ in the mesosphere and lower thermosphere comes from the $\Delta v_3$ and $\Delta v_1$ bands.



205  where chemical production has been neglected. In summary, for daytime, the required [O] is computed by using this equation and the $O_3$ retrieved in the previous iteration.

The photo-absorption coefficient $J_{O3}$ is calculated by using the TUV model version 5.3.2[3], which includes the MIP6 solar spectral irradiance. In the previous version we used TUV version 4.2 which uses the less variable SUSIM (Solar Ultraviolet Spectral Irradiance Monitor) solar spectral radiance. The new $J_{O3}$ coefficient is ∼10% smaller below 100 km, which leads to

210  a daytime [O] of ∼10% smaller below 90 km, and is very similar near 100 km. As explained above, the effect of this change in the non-LTE model tends to decrease the retrieved $O_3$.

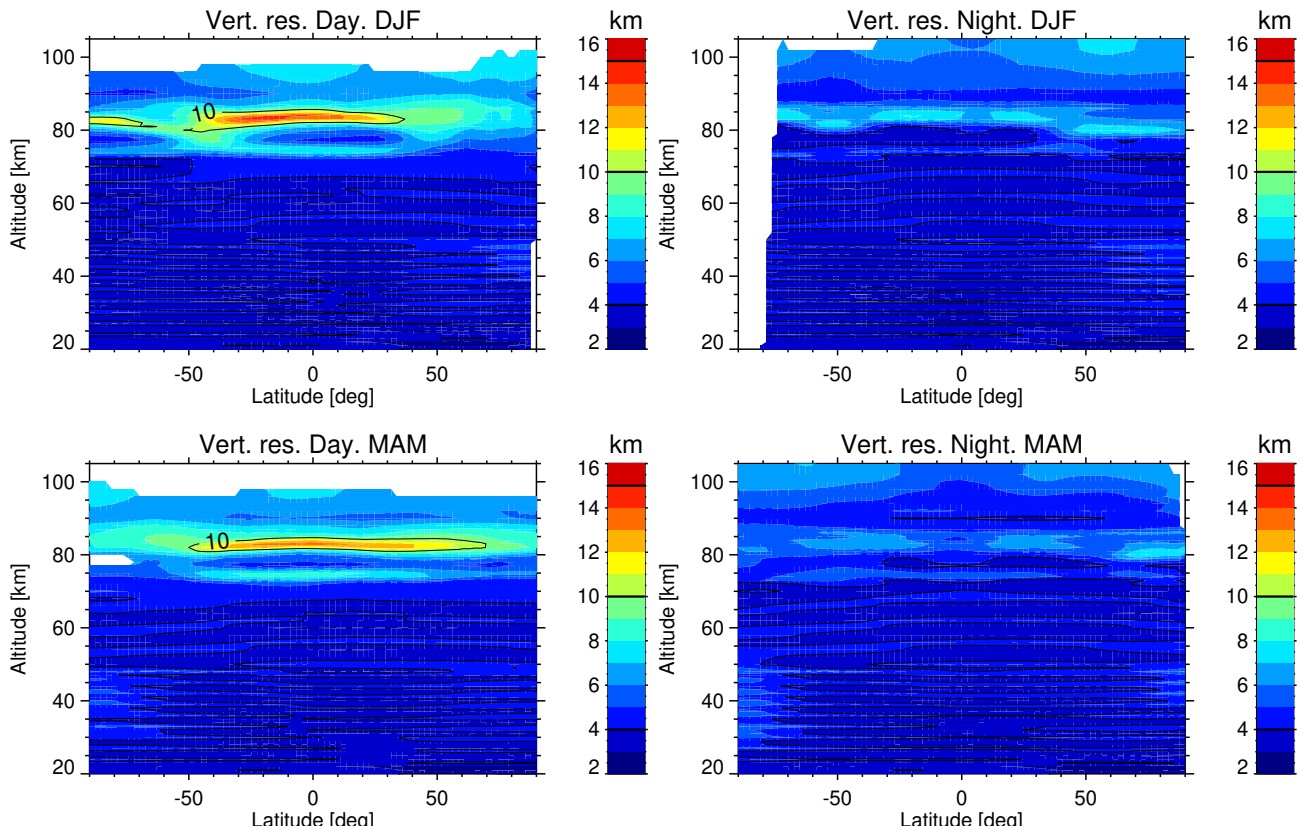

**Figure 1.** Latitude-altitude cross sections of MIPAS MA vertical resolution for daytime (left panels) and nighttime (right panels) conditions. Top panels are for solstice (December-January-February: DJF) and bottom panels for equinox (March-April-May; MAM). The means include all measurements from 2005 to 2012. White areas denote regions where the retrieved $O_3$ is not significant. Contour lines are marked in the color bar scale.

---

[3]https://www2.acom.ucar.edu/modeling/tropospheric-ultraviolet-and-visible-tuv-radiation-model.

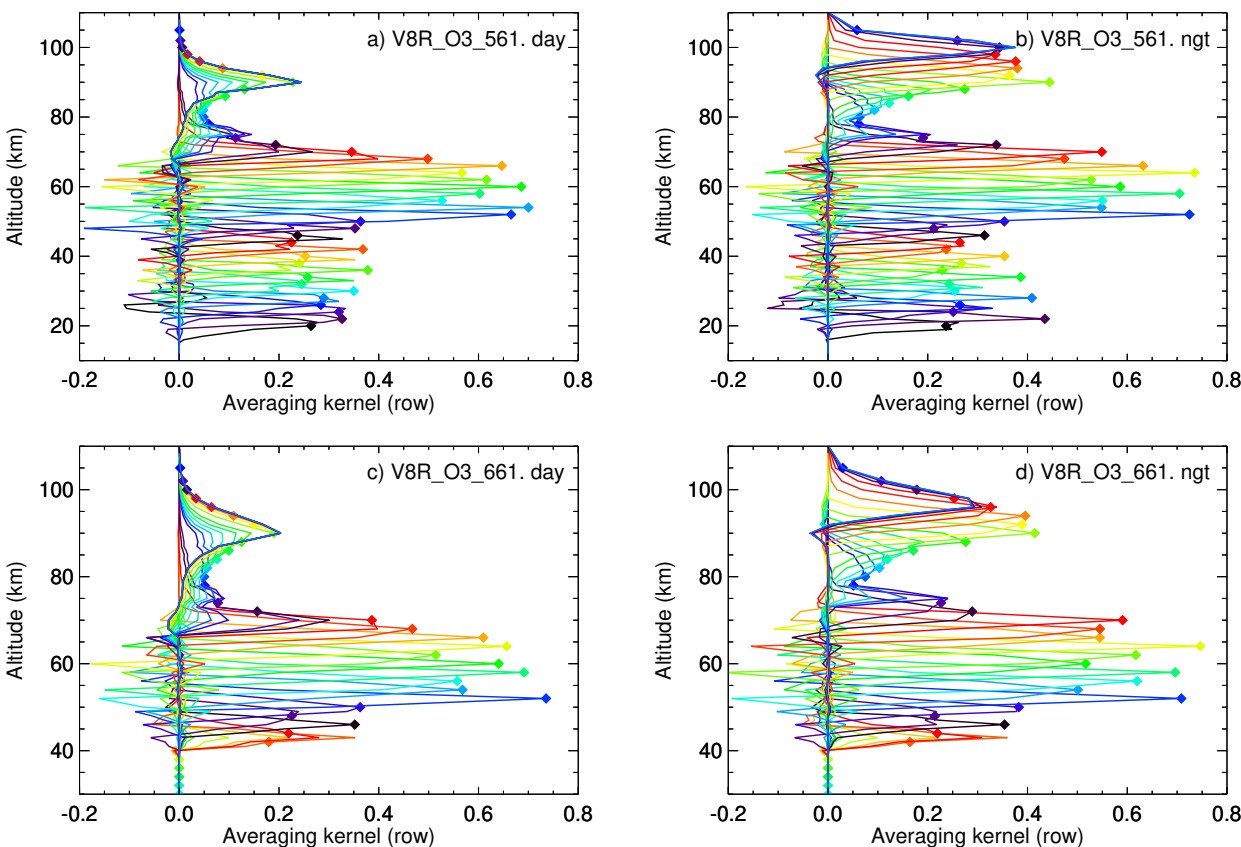

**Figure 2.** Rows of the averaging kernel matrix for ozone profiles recorded during the MA (top row) and UA (bottom row) measurement period. The corresponding averaging kernel diagonal elements are indicated by symbols. Left panels show the values for daytime conditions and the right panels for nighttime conditions. Kernel rows are shown every 2 km up to 102 km and at 115 km. MA data in a) are taken from 1 Sep 2009 (Envisat orbit 39234) at 49.8°N, 152.8°E, and in b) from 10 Sep 2009 (Envisat orbit 39364) at 42.0°N, 51.1°W. UA data in c) and d) are taken from 1 Sep 2009 (Envisat orbit 39235) at 49.8°N, 127.9°E, and at 40.5°N, 46.6°W, respectively.





**Nighttime**

During nighttime, the photochemical equilibrium of $O_3$ leads to

$$[O]_n \approx \frac{k_2[H]\,[O_3]}{k_1\,[O_2]\,[M] - k_3\,[O_3]}. \qquad (2)$$

In the previous version we calculated $[O]_n$ from Eq. (2) and taking $[H]$ from the NRLMSIS-00 model (Picone et al., 2002). Here $[H]$ was inferred from MIPAS daytime and nighttime $O_3$ measurements in the altitude range where the O vmr is free of photochemically-induced diurnal variations (above $\sim$80 km). In that region, the atomic oxygen vmr has no significant diurnal variations except those caused by tides, which we consider related by $O_{vmr,n} = O_{vmr,d} \times f_t$, where $f_t$ is a tidal correction factor. Thus, by combining Eqs. 1 and 2, using $f_t$, and defining the chemical losses for day- (10 am) $L_d = k_{1,d}\,[O_2]_d\,[M]_d - k_{3,d}\,[O_3]_d$, and night-time (10 pm) $L_n = k_{1,n}\,[O_2]_n\,[M]_n - k_{3,n}\,[O_3]_n$, we obtain

$$[H]_n = \frac{J_{O3}}{k_{2,n}}\,\frac{L_n}{L_d}\,\frac{O_{3,vmr,d}}{O_{3,vmr,n}}\,f_t, \qquad (3)$$

where we consider different densities and temperatures for daytime and nighttime. The tidal correction factor $f_t$ for atomic oxygen can be constructed by making use of the pressure difference $\Delta p$ between 10 am and 10 pm at a given potential temperature level by

$$f_t = 1 + \frac{O_{3,vmr,d}(p - \Delta p)}{O_{3,vmr,d}(p)}. \qquad (4)$$

Unfortunately, MIPAS did not measure day- and night-time $O_3$ at the same geolocations. This means that information on $[H]_n$ can be obtained from MIPAS only on a climatological basis, e.g., from $O_3$ vmr day- and nighttime climatologies. In order to incorporate the transient variability we employ SD-WACCM4 simulations (see Sec. 2.2.2) sampled at MIPAS geolocations, $[H_W]_n(\mathrm{geo})$, e.g.,

$$[H]_n(\mathrm{geo}) = [H_W]_n(\mathrm{geo})\,f_c \qquad (5)$$

where, in order to be consistent with the climatological MIPAS $[H]_n$, they were corrected by the climatological MIPAS to WACCM ratio

$$f_c = [H_{MIP}]_n(\mathrm{month, p, lat})/[H_W]_n(\mathrm{month, p, lat}). \qquad (6)$$

This method for generating the a priori $[H]$ has two caveats. First, SD-WACCM4 is free-running above the stratopause, resulting in deviations from the observed meteorology. However, larger perturbations of MLT dynamics caused by wave propagation from the lower atmosphere, which may have a noticeable impact on atomic hydrogen, are still represented reasonably well in comparison to a merely climatological a priori. Secondly, the correction factor $f_c$ is based mainly on MIPAS V5 $O_3$ data. It might change if derived from the V8 reprocessed ozone data, which would then raise the necessity for iteration. To sort out this we computed $f_c$ factors from the resulting V8 ozone data. Differences in the updated $f_c$ are generally smaller than 15%, except in the tropics, where deviations can be as large as 30%. The nighttime $[H]$ concentration derived from MIPAS V5R data in the 80–100 km altitude range agrees well, within a 20–25%, with the $[H]$ from the NRLMSIS 2.0 empirical model





(Emmert et al., 2021) for all latitudes and seasons, except in the tropics where the differences are within 20% and 50%, with [H] derived from MIPAS being smaller.

The atomic oxygen described above is available in the region where we retrieve $O_3$. Above approximately 97 km, we used the atomic oxygen from SD-WACCM4 simulations (see García-Comas et al., 2023).

Even if we tried to avoid $CO_2$ lines in the MW selection (see above), to be on the safe side, we also included the contribution of $CO_2$ lines in the forward model. The $CO_2$ bands are also considered in non-LTE. A detailed description of the $CO_2$ non-LTE model and all the required input parameters can be found in Jurado-Navarro et al. (2016).

## 250    3    Characterisation of the retrieved $O_3$ mixing ratios

Ozone is reliably retrieved from 20 km in the MA mode (40 km for the UA and NLC modes) up to ~95 km during illuminated conditions and up to ~105 km during dark conditions. Figures D1 and D2 show monthly climatologies of MIPAS V8 $O_3$ for day- (10 am) and night-time (10 pm) conditions respectively, from the mid-stratosphere up to the lower thermosphere. Their major features, including their vertical and latitude distributions and diurnal variations, are discussed in detail in López-Puertas
et al. (2018). The updated figures for version V8 are shown in Sec. D.

The zonal mean of the vertical resolution for the middle atmosphere (MA) mode for solstice and equinox conditions and for daytime and nighttime are shown in Figure 1. The vertical resolution of the retrieved ozone is given by the full width at half maximum of the averaging kernels rows. These are shown for four typical examples corresponding to MA and UA for day and nighttime in Figure 2. For daytime, the vertical resolution is about 3–4 km below 70 km, 6–8 km at 70–80 km, 8–10 km at 80–
90 km (coarser in tropical regions), and 5–7 km at the secondary maximum (90–100 km). For nighttime conditions, the vertical resolution is similar below 70 km, but it is better in the upper mesosphere and lower thermosphere. Overall, it is about 4-6 km at 70–100 km (except a narrow region near 80 km where it takes values of 6–8 km), ~4–5 km at the secondary maximum; and 6–8 km at 100–105 km. The vertical resolution has not changed significantly from the previous version. If anything, they are slightly better now above about 80 km, more markedly during nighttime conditions. The results for the UA and NLC modes
are very similar in the common retrieved region.

The criteria recommended for using the data are the same as in the previous version, which we recall here. First, the individual values of the retrieved profiles where the diagonal value (or the mean diagonal value when averaging) of the averaging kernel is less (in absolute value) than 0.03 should not be used (data with smaller values are considered non-trustful); and, secondly, the values corresponding to altitudes not sounded by MIPAS (e.g., below the lowermost tangent altitude), which are flagged by
the visibility flag, should not be used.

## 4    Error budget

The evaluation of the error budget of this version of ozone is based on the error estimation scheme for MIPAS version 8 data described by von Clarmann et al. (2022), which follows the TUNER (Towards Unified Error Reporting) recommendations. The

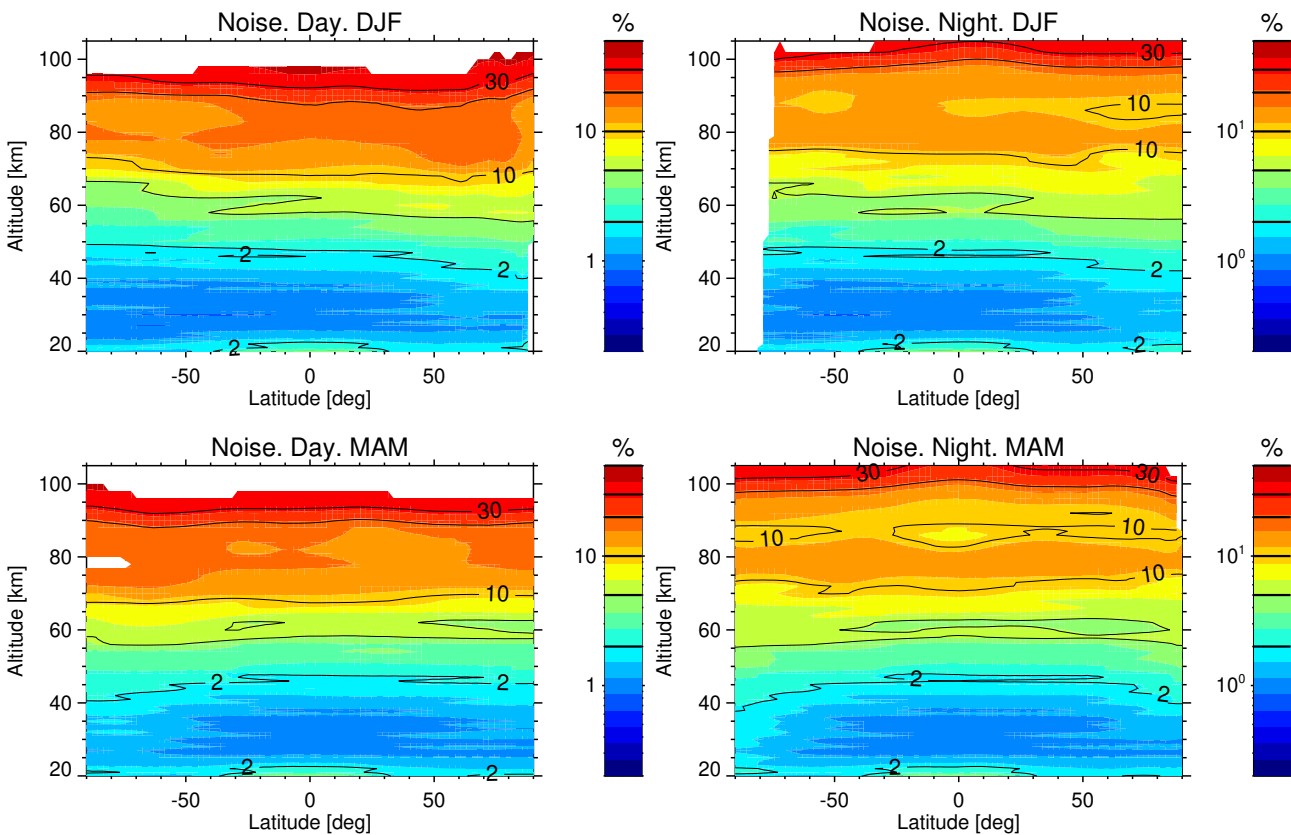

**Figure 3.** Latitude-altitude cross sections of the noise component of the ozone uncertainty in terms of estimated standard deviations for the MA mode. Left and right columns are for daytime and nighttime conditions, respectively. Top panels are for solstice conditions (Northern winter, December-January-February: DJF), and bottom panels for equinox conditions (Northern spring, March-April-May; MAM). The means span over all measurements from 2005 to 2012. White areas denote regions where the retrieved $O_3$ is not significant (AK diagonal <0.03). Contour lines are marked in the colour bar scale.

novelty (and improvement) of this scheme is that it allows accounting for error correlations which may result in error compensation. Further, it also allows the error propagation of uncertainties in parameters obtained in preceding retrievals. The errors are estimated on a single profile basis for the measurement noise and on representative atmospheric conditions for the rest of the errors. In particular, we considered 34 representative atmospheric conditions defined in terms of latitude band, season, and illumination conditions, which cover most of the climatologically expected situations for the middle atmosphere measurement mode (see Table A3 in Clarmann2022). The attribution of a given measured profile to these representative atmospheres is given in Table A6 of von Clarmann et al. (2022). For some species, like NO, CO and $CO_2$, we also have to consider the upper atmosphere measurement mode scenarios and even also different solar conditions (Tables A4 and A5 in von Clarmann et al., 2022). In the case of $O_3$, as its abundance is practically negligible above 105 km, the scenarios for UA are very similar to the





middle atmosphere one. Hence we will focus on the error estimates for the latter but the estimates for both UA scenarios are also included in the supplement information for completeness.

In the next sections, we discuss the major error sources and the corresponding uncertainties that we assume in the estimation of the $O_3$ errors. Following the TUNER recommendations given by von Clarmann et al. (2022) we discuss, separately, the errors thought to be of mainly random nature and those of systematic origin. The methodology of the error estimation for both types of errors is described in detail in von Clarmann et al. (2022). All estimated errors are given as standard deviations ($1\sigma$). A detailed analysis of the errors affecting the retrieved $O_3$ below about 70 km is already given in Kiefer et al. (2023). Nevertheless,

for completeness, we briefly describe them here and focus on the middle/upper mesosphere and lower thermosphere where non-LTE errors are very relevant.

## 4.1 Error sources and uncertainties

Again, based on the nomenclature of von Clarmann et al. (2022), we considered the following types of errors: measurement errors, parameter errors, and model errors. The measurement errors comprise essentially the measurement (spectral) noise and

additionally, those errors related to the instrument's state which are not well known (see Sec 4.1.1). Within the category of parameter errors fall the uncertainties of the atmospheric state parameters which are not known perfectly but not that much as being considered as unknowns of the retrieval (see Sec 4.1.2). In this type, we also include those which cannot be retrieved from the measurements because they do not contain enough information of them. Lastly, we consider as model errors the uncertainties in the spectroscopic data and the non-LTE parameters (see Secs. 4.1.3 and 4.1.4). Note that the latter is specific

to these atmospheric modes of observations and not included in the nominal mode.

### 4.1.1 Measurement errors

The measurement errors include the spectral noise of MIPAS, the uncertainty in the gain calibration, the instrument line shape uncertainty, inaccuracies in the frequency calibration and pointing errors.

    The propagation of the spectral noise was estimated by using Eq. 5 of von Clarmann et al. (2022). The mapping of the

other measurement errors on the $O_3$ error was evaluated from sensitivity studies performed for the representative atmospheric conditions discussed above.

    The measurement noise is typically $30-33\,\mathrm{nW/(cm^2\,sr\,cm^{-1})}$ in the MIPAS A band and $5.4-9.6\,\mathrm{nW/(cm^2\,sr\,cm^{-1})}$ in the AB band. Both values refer to the apodized spectra.

The mapping of those errors on the retrieved $O_3$ single profiles are shown in Fig. 3 for solstice conditions (top row) and for

equinox conditions (bottom row). As mesospheric $O_3$ is notoriously different during day and nighttime, and MIPAS sensitivity so is (see Fig. 2), we distinguish between daytime (left panels) and nighttime (right panels). Typical values ($1\sigma$) for daytime (left column of Fig. 3) are smaller than 5% below ~60 km, 5–10% between 60 and 70 km, 10–20% at 70–90 km and about 30% at 95 km. For nighttime (right column of Fig. 3), the $O_3$ noise errors are very similar to those during daytime below around 70 km but significantly smaller above, being 10–20% at 75–95 km, 20–30% at 95–100 km (except near the tropics where the

errors are smaller) and larger than 30% above 100 km.





The $1\sigma$ gain uncertainties were estimated to be 1.1% and 0.8% for the A and AB bands, respectively (see Kleinert et al., 2018). The response of retrieved ozone to the gain calibration is, to first order, multiplicative. Hence, in the band A spectral region, the gain uncertainties of the retrieved ozone mostly compensate for the gain uncertainty of the temperature and tangent altitude errors, since these are retrieved from the same band. This occurs approximately for the $O_3$ retrieved below about 50 km.

The instrument line shape (ILS) uncertainties and the spectral shift residual error have been evaluated as described in Kiefer et al. (2021) and are based on the estimates of modulation loss through self-apodization and its uncertainties. We also considered a residual frequency calibration error which resulted in $0.00029\,\mathrm{cm}^{-1}$ in bands A and AB.

### 4.1.2    Uncertainties in atmospheric parameters

In the ozone retrieval of the MA, UA and NLC modes we used the information of temperature and tangent altitude previously
retrieved from the $15\,\mu\mathrm{m}$ region. Thus, the different sources of errors affecting the retrieved temperature and tangent altitude (T-LOS) are implicitly taken into account by propagating their uncertainties into the $O_3$ retrieval. The major sources of temperature are the following. The random component, mainly instrumental noise, is typically less than 1 K below 60 km, 1–3 K at 60–70 km, 3–5 K at 70–90 km, 6–8 K at 90–100 km, 8–12 K at 100–105 km and 12–20 K at 105–115 km (García-Comas et al., 2023). Its systematic part is dominated by the $CO_2$ spectroscopic data errors below 75 km and by the uncertainties in the non-
LTE model parameters and $CO_2$ concentration above ∼80 km. The systematic uncertainties are smaller than 0.7 K below 55 km, 1 K at 60–80 km, 1–2 K at 80–90 km, 3 K at 95 km, 6–8 K at 100 km, 10–20 K at 105 km and 20–30 K at 115 km (García-Comas et al., 2023).

The uncertainties of the spectrally interfering molecules with ozone which are not jointly fitted (e.g. as in the case of water vapour), as well as their vertical covariances, are estimated from the error covariance matrices of previous MIPAS data version
V5. These values are considered more accurate than the climatological mean values for the actual atmospheric conditions of the measurements. For those interfering species which are not available from previous MIPAS versions we used climatological data and estimate the effects of their errors on the retrieved $O_3$ from the perturbed spectra with their $1\sigma$ estimated uncertainties. We should note that, overall, the errors in $O_3$ due to the uncertainties in the interfering species are very small. This is due to the high spectral resolution of MIPAS which allows us to choose MWs where essentially only $O_3$ contributes to the measured
radiance. The errors caused by the uncertainties in $H_2O$ are not explicitly included in the error budget as $H_2O$ is jointly fitted with $O_3$ and $H_2O$ uncertainties are implicitly taken into account during the propagation of measurement errors. Further, $CO_2$ uncertainties contribute to the $O_3$ error budget not only because of spectral $CO_2$ interferences (for which we also include $CO_2$ levels in non-LTE, see Sec. 2.2.4), but mainly through its propagation via the preceding temperature retrieval from band A (see von Clarmann et al., 2022). $CO_2$ was taken from WACCM4 simulations and their $1\sigma$ uncertainties are listed in Table 3 of
Kiefer et al. (2021).

In Sec. 2.2.4 we discuss the concentrations of the atomic oxygen and atomic hydrogen, which are not measured by MIPAS but required for the calculations of the non-LTE population of the $O_3$ levels. The atomic oxygen uncertainty contributes to the $O_3$ error budget in two ways. One, directly, through the non-LTE population of the emitting levels, as described above; and the other, indirectly through its impact on the kinetic temperature retrieval (see García-Comas et al., 2023). It is worthwhile to note





that both errors partially compensate when propagated to the retrieved $O_3$. Thus, in the upper mesosphere, a larger [O] induces a larger retrieved kinetic temperature and hence a larger population of the $O_3$ emitting levels. On the other hand, a larger [O] gives rise to a larger collisional deactivation of the $O_3(v_1, v_3)$ levels and hence lower populations. In the lower thermosphere, the effects are swapped but with a similar compensating effect.

As described above the daytime atomic oxygen is obtained from the retrieved $O_3$ below 95 km. It is then inherently retrieved

together with $O_3$ and it is not by itself an independent source of errors. The uncertainties incurred by this approach are those introduced by the collisional and kinetic rates, which are described below in Sec. 4.1.4. During the nighttime, the [O] uncertainty is driven by that of atomic hydrogen. The uncertainties in [H] have been estimated by using Eq. (2) and the errors of the entering parameters which have been discussed above. They result in mean values ranging between 20 and 40% from 70 to 100 km, being larger in the tropics (40–60%) and smaller at high latitudes (10–30%).

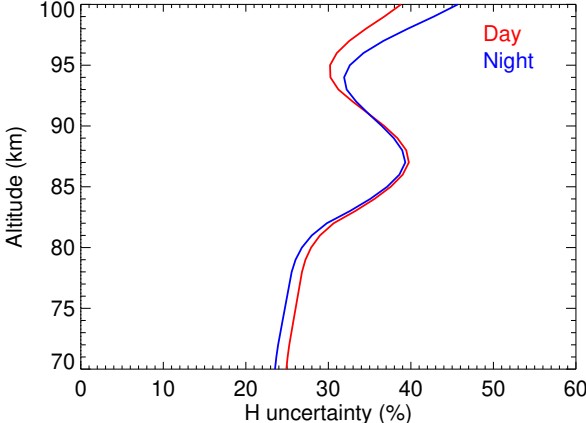

**Figure 4.** Mean of the [H] uncertainties included in the $O_3$ error budget for day and nighttime conditions.

In the region of 95–120 km, where [O] is taken from WACCM, we use the same uncertainties (5–30%) as those used in the kinetic temperature retrieval (see Fig. 6 in García-Comas et al. (2023)).

The "indirect" error induced by O through the temperature retrieval has been neglected. First, it is partially compensated by its direct effect on the populations of $O_3$. Secondly, the effect of the [O] error on the temperature uncertainty is much smaller than those of other parameters (e.g. of $k_{CO2-O}$ (García-Comas et al., 2023)) already included. Further, we should note that

the propagation of the temperature errors onto $O_3$ is considerably weakened in the mesosphere since the emitting levels are in non-LTE, e.g., far away from the population dictated by the local kinetic temperature.

### 4.1.3   Uncertainties in spectroscopy

The spectroscopic errors are in general not so well characterised as it was required for a thorough error assessment of remotely sensed data. In particular, neither any information on the confidence limits of the error margins nor about error covariances

between spectral lines and bands is available. The spectroscopic errors for $O_3$ are described in detail by Kiefer et al. (2023); see



their Table 2. In summary, for the stronger ozone lines, we assume uncertainties in their intensities in the range of 2–4%. The broadening coefficients were parameterised as a function of the rotational quantum number and their uncertainties vary from 3.5%, 7.5%, 15%, to 20%, depending on the considered ozone band. In this respect, we were rather conservative and assumed that all errors in the lines of a given band are completely correlated.

### 4.1.4 Uncertainties in non-LTE parameters

The uncertainties in the non-LTE model are an important source of systematic error for $O_3$ above the mid-mesosphere: On one hand, the uncertainties in the collisional rates and, on the other, the atomic oxygen concentration, which has been discussed above. The errors of the collisional rates were discussed in López-Puertas et al. (2018) and their values have not changed in this version. In summary, we considered an uncertainty of: 1) 10% for the three-body reaction rate of $O_3$ formation ($k_1$ in Table 1); 2) 20% for the thermal relaxation of $O_3(v_1, v_3)$ by $N_2$ and $O_2$; and 3) 50% for the collisional relaxation (and/or chemical reaction) of $O_3(v_1, v_3)$ with atomic oxygen ($k_3$ in Table 1), the latter based on West et al. (1976, 1978). The error of the photo-absorption coefficient $J_{O3}$, requires in the photochemical model described above, was estimated in 7%.

### 4.2 $O_3$ error estimates

While in the preceding section, ingoing uncertainties affecting the retrieval were reported, we here assess their effect on retrieved ozone mixing ratios. As discussed above we distinguish between the errors which are mainly random and those of a major systematic nature. All error components, of one or another nature, contributing to the $O_3$ error budget for the considered 34 atmospheric conditions are given in the supplement document. That information includes tables as well as figures for the MA (Tables S2–S35 and Figs. S1–34) and for the UA (Tables S37–S70 and Figs. S35–S68) measurement modes. We show in Fig. 5 three examples for the middle atmosphere (MA) mode corresponding to the tropics, to northern mid-latitudes and for polar summer and polar winter conditions. We showed figures for the two illumination conditions because mesospheric ozone is very different during day and nighttime. The numeric values at selected altitudes for the northern mid-latitude spring (day and night) are given in Tables B1–B6 for easier reference. They also list mean $O_3$ vmrs which can help in estimating absolute $O_3$ errors.

Many error components cause both a bias and scatter and thus contribute both to the random and the systematic error budget. In the figures, the combined random and systematic error is shown for each error source. Both, propagated measurement noise and noise-induced T-LOS errors as considered as purely random. Some error sources affect the ozone retrieval via multiple pathways, first via their propagation onto T-LOS, which is further propagated on $O_3$, and second directly. For example, too large radiances due to less-than-perfect gain calibration lead to high temperatures that cause lower retrieved $O_3$ mixing ratios. Along the second pathway, too large radiances in the $O_3$ band cause also higher retrieved $O_3$ mixing ratios. In the figures we show the net effect of both pathways, labelled by the respective error source, while the error component labelled 'T-LOS' contains only the estimated noise-induced T-LOS error. In the next sections we assign the error components to the random versus systematic category and discuss their relevance. An exception is non-LTE errors, which are very relevant in the mesosphere and above and are discussed in a separate section, 4.2.3, describing its two components together.





**Figure 5.** O$_3$ error budget for MA data for different atmospheric conditions (top to bottom) tropics, northern mid-latitudes in spring and polar regions, for daytime (left column) and nighttime (right column). All error estimates are 1σ uncertainties and are given in percent. Error contributions are labelled "T+LOS" for the propagated error from the T+LOS retrieval, "noise" for error due to measurement noise, "spectro" for the spectroscopic error, "gain" for gain calibration error (see text), "offset" for error due to spectral offset (see text), "ILS" for instrument line shape error (see text), "interf" for the uncertainty in the abundance of the interfering species, and "NLTE" for non-LTE related errors. The total random ("random") and systematic ("syst") errors are also shown.



### 4.2.1 Random errors

In line with the recommendation of TUNER, we consider random errors those that explain the standard deviation of the differences of collocated measurements taken by two instruments of the same state variable (von Clarmann et al., 2022).

The following error sources are considered to contribute to the random errors: measurement noise, gain calibration uncertainties, offset calibration noise, T-LOS errors, the uncertainties in the abundance of interfering species, and the residual frequency calibration errors. Further, we also include in the random error the random variations of retrieval responses to systematic uncertainties (so-called "headache errors" (see von Clarmann et al., 2022).

The random component of the errors in Fig. 5 are shown in Fig. C1. The major component of the random error is the measurement noise. Zonal mean distributions of the noise error for solstice and equinox, and for daytime and nighttime, have been discussed above in Sec. 4.1.1 and are shown in Fig. 3. The noise error (+) and the total random error (thick red line) (see Fig. 5) are practically overlaid above ∼70 km. Above ∼80 km, the offset, the random component of the NLTE and, in some cases, the random component of the TLOS, are the major components after the measurement noise. Between ∼70 km and ∼50 km, the propagation of the temperature and LOS random errors significantly contributes. Below ∼50 km, both the T-LOS and the random component of the spectroscopic errors contribute significantly to the total random noise.

The random component of the gain calibration error is very small and the values shown in Fig. 5 are mainly of systematic origin. The errors induced by the uncertainties in the abundance of interfering species are significant only below about ∼30 km; and those produced by the residual frequency calibration are negligible and are not shown.

The total random error varies between ∼1–2% near 30 km, and increases with altitude reaching 20%–30% in the upper mesosphere/lower thermosphere. In this region, it is larger at daytime than at nighttime. In the lower stratosphere, due to the temperature decrease, it also increases (see more details in Kiefer et al., 2023). The random $O_3$ error is generally larger than the total systematic uncertainty above around ∼60 km.

### 4.2.2 Systematic errors

The sources of systematic errors in MIPAS $O_3$ retrieval are uncertainties in spectroscopic data, instrument line shape uncertainties, the persistent component of gain calibration error, and non-LTE related uncertainties. The systematic component of the errors in Fig. 5 are shown in Fig. C2. The systematic error is dominated by the spectroscopic data error below ∼60 km. In this region, the ILS also contributes significantly and, to a lesser extent, the persistent part of the gain calibration uncertainty. Above ∼60 km the systematic errors are dominated by the NLTE uncertainties (which are discussed below) but the spectroscopic data error also contributes significantly being in some cases comparable or even larger than the NLTE errors. The ILS contribution is also appreciable at very high altitudes, above ∼90 km.

### 4.2.3 Non-LTE errors

The non-LTE error includes both uncertainties of the collisional and kinetic rate constants, and uncertainties of atmospheric abundances required for the non-LTE modelling (O and H). For the [O] and [H] errors, we expect that the systematic uncertainty







**Figure 6.** O$_3$ NLTE error budget for MA data for different atmospheric conditions (top to bottom) tropics, northern mid-latitudes in spring and polar regions, for daytime (left column) and nighttime (right column). All error estimates are 1$\sigma$ uncertainties and given in percent. Error contributions are labelled as: 'H' the error due to [H], 'O+O$_2$+M' the error due to $k_1$, 'O$_3$-M' the error of the collisional relaxation of O$_3(v_1,v_3)$ by M (N$_2$ and O$_2$), 'O3-O' the error of the collisional relaxation/chemical reaction of O$_3(v_1,v_3)$ with O ($k_3$), '$J_{O3}$' the error of the photo-absorption coefficient $J_{O3}$, 'O-error' the error of [O] above ~95 km, and "NLTE" is the total non-LTE contribution.





component, either caused by the uncertainties in the kinetic constants of the photochemical model or by biases in the considered climatologies, is more relevant than the variability counterpart. Only in a few conditions and altitudes the random component of the non-LTE is significant, e.g., near 95 km in daytime in the tropics and in the northern polar summer (see Fig. 5).

The different components of the non-LTE errors corresponding to the atmospheric conditions of Fig. 5 are shown in Fig. 6.

Non-LTE errors are only significant only above $\sim$70 km and are comprised in the range of 1% to 10%. In the region of 65–85 km the non-LTE error is dominated by the uncertainty in the thermal relaxation of $O_3(v_1, v_3)$ by $N_2$ and $O_2$, and above $\sim$85 km by the errors in the collisional relaxation/chemical reaction of $O_3(v_1, v_3)$ with atomic oxygen and in [H]. The latter is the largest contributor above $\sim$90 km during nighttime conditions at tropical and mid-latitudes. The error component due to O in the region where it is not retrieved (z$\gtrsim$95 km) contributes moderately at the highest altitudes. The error contribution of

the reaction rate of $O_3$ formation ($k_1$ in Table 1) is negligible during nighttime and below 2% during daytime. The error due to the photo-absorption coefficient is hardly 1% near 80 km during daytime. Overall, the non-LTE errors are typically negligible (smaller than 1%) below 60 km, 2–8% at 60–85 km, and slightly larger, 5–12% above 85 km.

## 5  Differences between current V8R_O3_m61 and previous V5R_O3_m22 $O_3$ data

The average impact on the ozone retrieval after including the changes discussed above is an increase of 2–5% (0.2–0.5 ppmv)

below around 50 km (see Figs. 7 and 8a). Also, there is a clear decrease by $\sim$2–4% between 50 and 60 km, mainly in the tropics and mid-latitudes (see Fig. 8b). In the region between 70 and 85 km we observe an oscillating behaviour of the differences with amplitudes smaller than $\sim$10%. The concentration of $O_3$ in these altitudes is very small (below $\sim$0.5 ppmv); hence these differences are not of much importance in absolute terms (see left panels of Fig. 7).

The position and magnitude of the $O_3$ tertiary maximum, located near 70 km at high winter latitudes, are very similar in both

versions, with no significant difference. For example, compare panels (d) and (f) in Fig. D5 with the corresponding panels in Fig. 15 of López-Puertas et al. (2018).

The nighttime $O_3$ minimum just below 80 km is more pronounced in the new version. This is clearly seen in the right/bottom panel of Fig. 7 and in Fig. 8c; compare also panel (b) in Fig. D5 with the corresponding panel in Fig. 15 of López-Puertas et al. (2018). Nevertheless, it is interesting to note that the $O_3$ fields near 78 km show larger values at daytime than at nighttime

(see Figs. 8c and D3). This day/night difference is of a different sign than in the previous version (see Fig. 13 in López-Puertas et al., 2018) but it is in agreement with both, SABER measurements and the WACCM predictions (see López-Puertas et al., 2017). Hence, the new $O_3$ dataset in this region seems to have improved and shows a more realistic diurnal variation in the mesosphere.

The $O_3$ concentration in the secondary maximum during daytime in the tropic and midlatitudes is about 40% larger in the

new version compared to the previous version (see top/right panel of Fig. 7 and Fig. 8d). In absolute terms, however it is just 0.2–0.3 ppmv. The daytime secondary maximum is also about 1–2 km higher than in the previous version (e.g. compare panel (a) in Fig. D5 with the same panel in Fig. 15 of López-Puertas et al., 2018). The reason for these differences is the use of a different a priori $O_3$, the WACCM-corrected fields instead of the values of the Garcia and Solomon (1994) model, together with





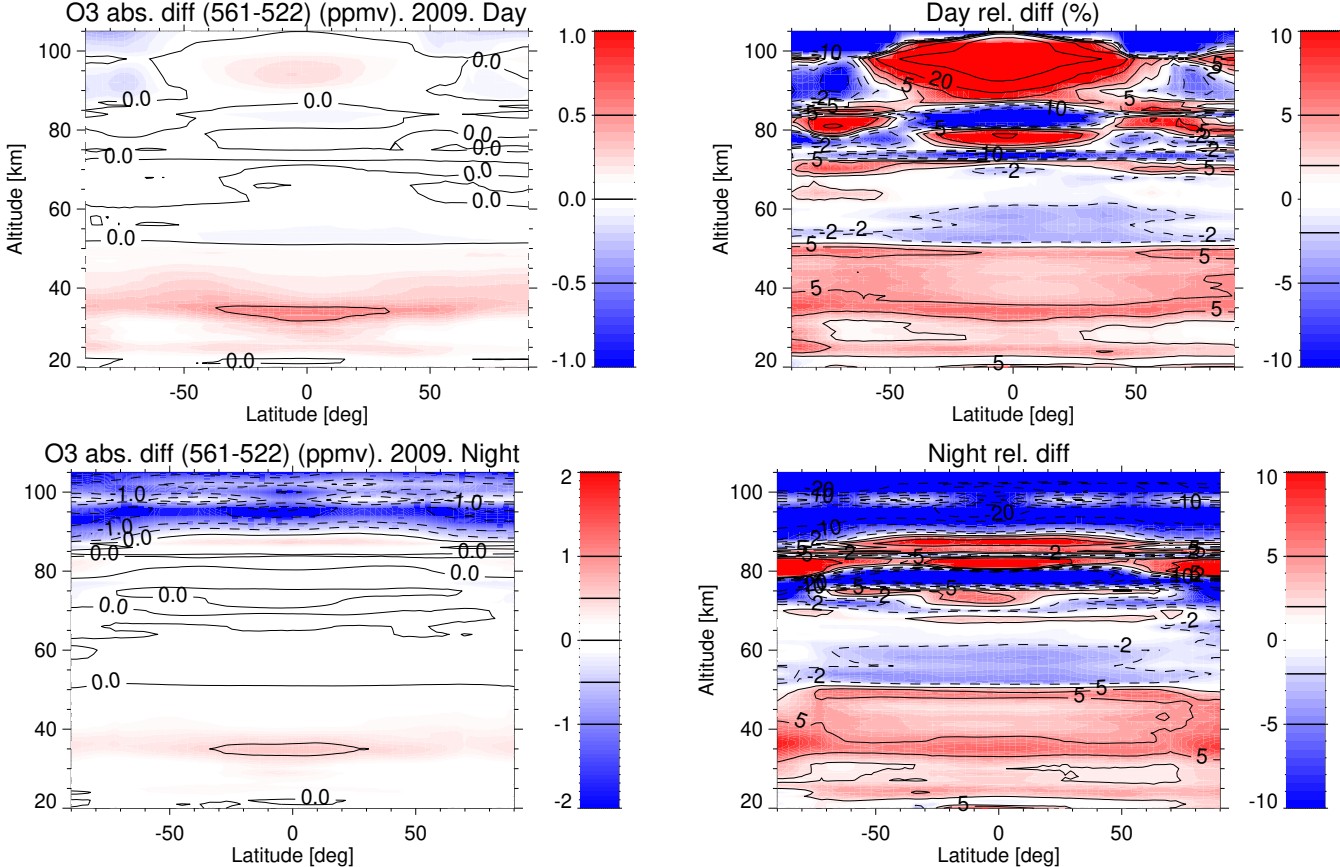

**Figure 7.** Comparison of $O_3$ abundance retrieved in the V8 561 with the previous V5R_O3_m22 version for daytime conditions (10 am) (top) and nighttime (bottom). The plots show the mean of the differences, in % of the older version, (V8–V5R_O3_m22)/V5R_O3_m22, for all data taken in 2009.

the relatively low vertical resolution of MIPAS in this region for daytime (see left panels of Fig. 1). This effect overcomes the

smaller $O_3$ values expected from the lower photo-absorption coefficient $J_{O3}$ and the lower de-activation rate of the $O_3(v_1,v_3)$ states by atomic oxygen (see Sec. 2)

Probably the most significant change in the mesosphere occurs at nighttime in the secondary maximum, where $O_3$ in the new version is reduced between 10–30% (see bottom/right panel in Fig. 7 and Fig. 8d). This is caused mainly by the lower collisional relaxation/chemical removal rate of the $O_3(v_1,v_3)$ states by atomic oxygen, a factor of two smaller (see Sec. 2.2.3). This feature

can also be appreciated very clearly when comparing Fig. D2 and Fig. 12 in López-Puertas et al. (2018), particularly near the tropics during equinox conditions (when the maximum of $O_3$ occurs). Further, it is also evident when comparing panels (b) in Fig. D5 and in Fig. 15 of López-Puertas et al. (2018).

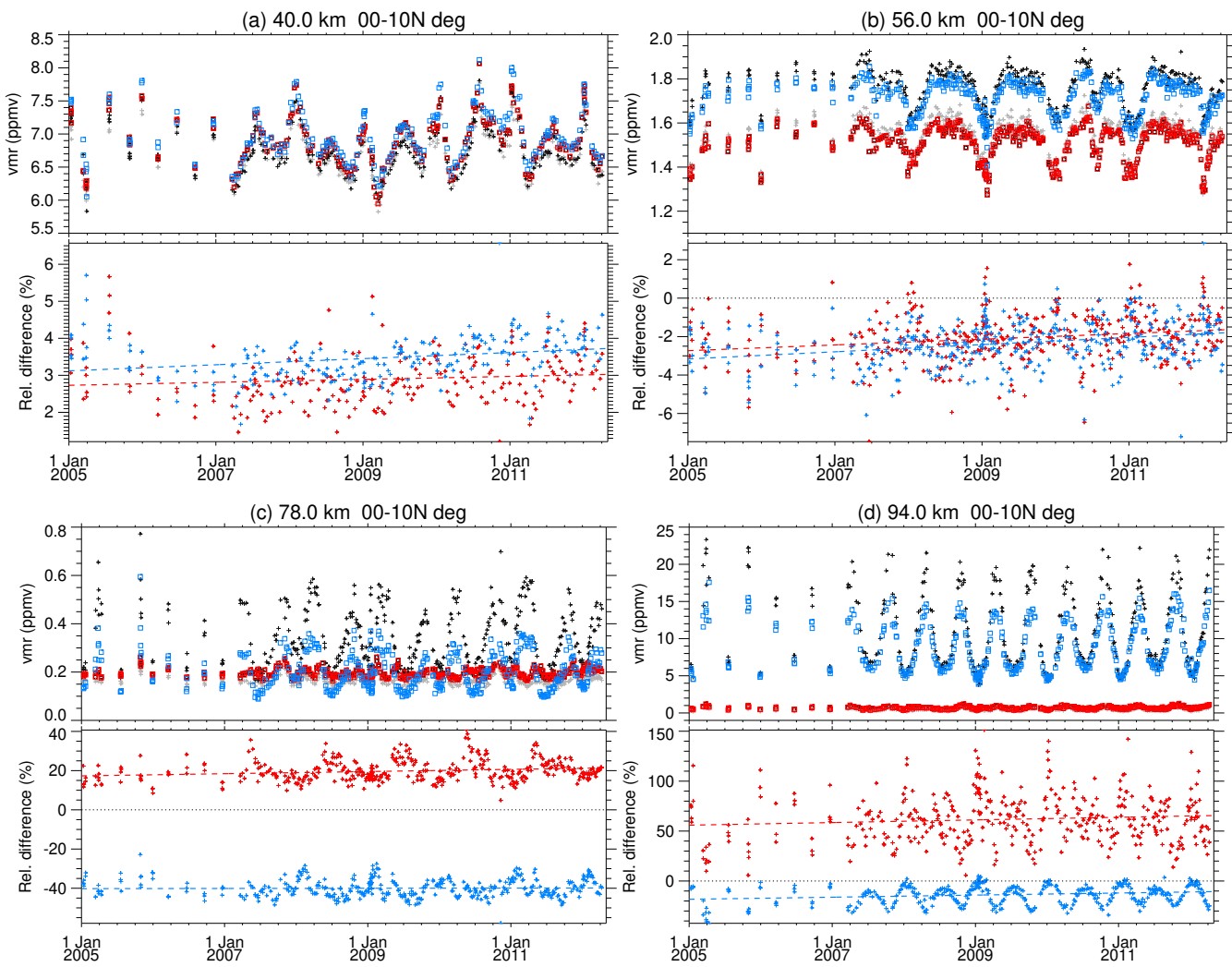

**Figure 8.** Comparison of $O_3$ abundance retrieved in the V8R_O3_561 version with that in the previous V5R_O3_m22 version for daytime (10 am) (red, light pluses) and nighttime (blue, dark pluses). Squares: V8R_O3_561; pluses: V5R_O3_m22. Top and bottom panels show the time series of $O_3$ vmr and of the relative differences, (V8-V5)/V5, respectively. Figures a–d are for 40 km, 56 km, 78 km, and 94 km, respectively, all for 0–10° N.



**Figure 9.** Comparison of O$_3$ zonal mean abundances retrieved in the V8 MA and UA data with the V8 NOM data for four seasons (rows). From top to bottom, for the Northern winter season (December–February, DJF), for the Northern spring (March-May, MAM), for the Northern summer winter (June-August: JJA) and for Northern autumn (September-November, SON). The left panels show the zonal mean of V8 MA/UA data, including 10 am and 10 pm. The central and right panels show the NOM–MA/UA differences for daytime (10 am) and nighttime (10 pm). The averages cover the measurements taken from 2006 through 2011.





# 6 Consistency between nominal and middle atmosphere O$_3$ data

Data merging of diverse series of data is nowadays a common procedure for obtaining long-term datasets and studying possible

changes and trends. In that sense, the NOM and the MA/UA O$_3$ MIPAS datasets might be used in the future for those purposes. Hence, it is important to know the consistency between the two datasets. NOM has the advantages that were measured more frequently, though limited in altitude range and including non-LTE (expected to be important in the 50–70 km range) in an approximate way only (Kiefer et al., 2023). The MA/UA V8 dataset described here is however rather discontinuous in time, approximately one out of ten days in each mode. Differences between both datasets are shown in Fig. 9 for four seasons

and separately for 10 am and 10 pm. The general pattern is that both datasets seem very consistent (not in vain we tried to use as close as possible retrieval setups). Differences are generally within 1–2%. The larger differences appear in the lower mesosphere (60–70 km) during daytime where NOM O$_3$ seems to be generally larger than MA/UA in the range of 2% to 10% for most latitudes. These differences are probably caused by a more accurate account of non-LTE, known to be more pronounced in the daytime. One clear exception to that general feature is the tertiary maximum occurring near 60-70 km in the

polar winter conditions (near the North pole during DJF and close to the South pole in JJA). In those regions, O$_3$ MA/UA is larger in ∼5–10%. A similar case occurs near the South pole for MAM, where O$_3$ is also enhanced. A possible cause for these differences could be that NOM retrievals are not including appropriately the O$_3$ field above its uppermost tangent height of the measured spectra (∼70 km). There are also other patches at high latitudes and at different altitudes where the NOM O$_3$ is slightly larger than MA/UA O$_3$ in 2–3%, but these are small and do not appear to be systematic.

# 7 Comparison with other satellite measurements


We have compared MIPAS retrievals of this new version of O$_3$, V8R_O3_m61, with co-located measurements from SABER, GOMOS, MLS, SMILES and ACE-FTS; in a similar way as we did for the previous version (López-Puertas et al., 2018). We have used more recent versions for the cases of MLS and ACE-FTS.

Comparisons for GOMOS are only for night conditions and are performed in number density. For ACE-FTS, because it is

an occultation instrument and O$_3$ has very large diurnal variations around the terminator in the middle and upper mesosphere, we compare ACE sunset and sunrise with MIPAS observations taken at solar zenith angles (SZA) from 88° to 92°. We kept the same co-location criteria as in the previous version, e.g., we selected pairs of profiles of the different instruments with Universal Time differences smaller than 2 h and distances smaller than 1000 km.

## 7.1 Instruments

### 7.1.1 SABER

SABER (Sounding of the Atmosphere using Broadband Emission Radiometry) is a broadband radiometer flying onboard NASA's Thermosphere-Ionosphere-Mesosphere Energetics and Dynamics (TIMED) satellite, launched on December 2001 and starting operations in January 2002 (Russell III et al., 1999). It measures from 83°S to 52°N and from 52°S to 83°N,





alternatively every two months. A 24-h local time coverage is completed in ∼60 days. The instrument measures the ozone
limb emission at 9.6 μm during daytime and nighttime, and the ozone concentration is retrieved from 10 to 100 km using a
non-LTE model (Mlynczak et al., 2013). Here we use version 2.0 of $O_3$ retrieved from the 9.6 μm channel, publicly available
at http://saber.gats-inc.com. SABER's ozone precision is $\approx 1-2\%$ in the stratosphere and $\approx 3-5\%$ in the lower mesosphere
(Rong et al., 2009). The systematic errors range from 22% in the lower stratosphere to $\approx 10\%$ in the lower mesosphere. The
vertical resolution of SABER ozone is approximately 2 km. Given MIPAS $O_3$ coarser vertical resolution, particularly in the
daytime mesosphere, we used the MIPAS averaging kernels and a priori $O_3$ to smooth SABER $O_3$ profiles.

### 7.1.2 GOMOS

The Global Ozone Monitoring by Occultation of Stars (GOMOS) was a stellar occultation spectrometer onboard the ESA's
Envisat space platform (Bertaux et al., 2010) and operated from August 2002 to April 2012. GOMOS took measurements at
$250-692$ nm and $O_3$ nighttime density profiles were derived from 10 to 110 km. The latitudinal coverage is uneven providing
data at low latitudes at around $22-23$h LST and eventually reaching the poles and slightly varying throughout the year. We
used here the ESA IPF version 6.01 which is described in Kyrölä et al. (2010) and Sofieva et al. (2010) and available from the
ESA Earth online portal (https://earth.esa.int). Unreliable profiles have been removed following the recommendations of the
GOMOS/6.01 Level 2 Product Quality Readme file.

GOMOS $O_3$ vertical resolution changes from 2 km in the lower stratosphere to 3 km in the upper stratosphere and above.
Because this resolution is better than that of MIPAS, we applied MIPAS averaging kernels to the GOMOS profiles. Because
GOMOS provides $O_3$ number density we compare the GOMOS and MIPAS $O_3$ number densities. Random errors due to
measurement noise and scintillations are $0.5-4\%$ in the stratosphere and $2-10\%$ in the mesosphere. Systematic errors are
smaller than 2%, mainly induced by the $O_3$ spectroscopic data (Tamminen et al., 2010).

### 7.1.3 MLS

The Microwave Limb Sounder (MLS) was launched on July 2004 on the NASA's Earth Observing System Aura satellite
(Waters et al., 2006). The ozone dataset used here is version 5.0-1.1a, downloaded from GES DISC (Schwartz et al., 2015)
and described by Livesey et al. (2022). One of the major differences of this data version is that the ozone retrieval uses larger
a priori errors at high altitudes, making the retrieved $O_3$ less dependent on the a priori so it can be used up to higher altitudes,
to 0.002 to 0.001 hP (∼90 km). Thus, we used here daytime and nighttime $O_3$ profiles from the stratosphere up to 0.001 hPa
(∼90 km (see https://mls.jpl.nasa.gov/products/o3_product.php). This is an important difference with respect to the previous
comparison carried by López-Puertas et al. (2018), where it was limited to ∼72 km.

The vertical resolution is 3 km in the stratosphere, 6 km in the middle mesosphere and 9 km in the upper mesosphere. The
estimated systematic uncertainty (accuracy) is 5–10% in the stratosphere, 10–20% in the lower mesosphere and 20–40% in the
middle mesosphere (with a larger error of 100% near 80 km) (see Table 3.18.2 in Livesey et al., 2022). Since MLS $O_3$ vertical
resolution in the mesosphere is larger than that of MIPAS, we have applied MLS averaging kernels and a priori information to
the MIPAS ozone.



**Figure 10.** Mean of the daytime $O_3$ vmr differences (MIPAS–instrument) in % of MIPAS between co-located pairs of measurements of MIPAS (MA mode) with ACE-FTS (green), MLS (purple), SMILES (magenta) and SABER (red) for a) spring (MAM for NH and SON for SH), b) autumn (SON for NH and MAM for SH), c) summer (JJA for NH and DJF for SH), and d) winter (DJF for NH and JJA for SH). The symbols indicate the mean altitude of the MIPAS $O_3$ vmr primary (diamonds) and secondary (circles) maxima coincident with the respective instrument. The number of coincidences is indicated in the subscripts. The colour-shaded areas (hardly noticeable in many cases) are the standard errors of the mean of the differences. The grey shaded area shows the MIPAS $1\sigma$ systematic errors (e.g. curve 'syst' in Fig. C2).



**Figure 11.** As Fig. 10 but for nighttime $O_3$. Instruments colours are the same except that ACE-FTS is replaced by GOMOS (light blue).

### 7.1.4 SMILES

The Superconducting Submillimeter-Wave Limb-Emission Sounder (SMILES) on the International Space Station (ISS) oper-
ated between October 2009 and April 2010 (Kikuchi et al., 2010). It measured ozone profiles in the interval of 16 to 85 km
545 during daytime and to 96 km during nighttime (Mitsuda et al., 2011; Takahashi et al., 2010, 2011). The latitudinal coverage
is 38°S and 65°N. Data version 3.2 was used here (http://darts.isas.jaxa.jp/stp/smiles/). The vertical resolution is 3 km in the
stratosphere, 4 km in the lower and mid-mesosphere and 6 km at 95 km. MIPAS and SMILES $O_3$ vertical resolutions are similar
and hence no averaging kernels was applied. Previous versions of SMILES $O_3$ (v2.2) agree with other measurements within
10% in the stratosphere and 30% in the mesosphere (Imai et al., 2013a,b).





### 7.1.5 ACE-FTS

The Fourier Transform Spectrometer (ACE-FTS) is an infrared solar occultation Michelson interferometer flying on the CSA's Atmospheric Chemistry Experiment (ACE), launched in August 2003 (Bernath, 2017). It measures atmospheric absorption from the cloud top to 150 km at sunrise and sunset. ACE covers the tropical, mid-latitude and high-latitude regions in approximately three months. Ozone profiles are retrieved from microwindows between 829 and 2673 cm$^{-1}$ (Boone et al., 2013). We use here data version 4.1. The retrievals are limited to 5–95 km altitude range. The vertical resolution is $3-4$ km.

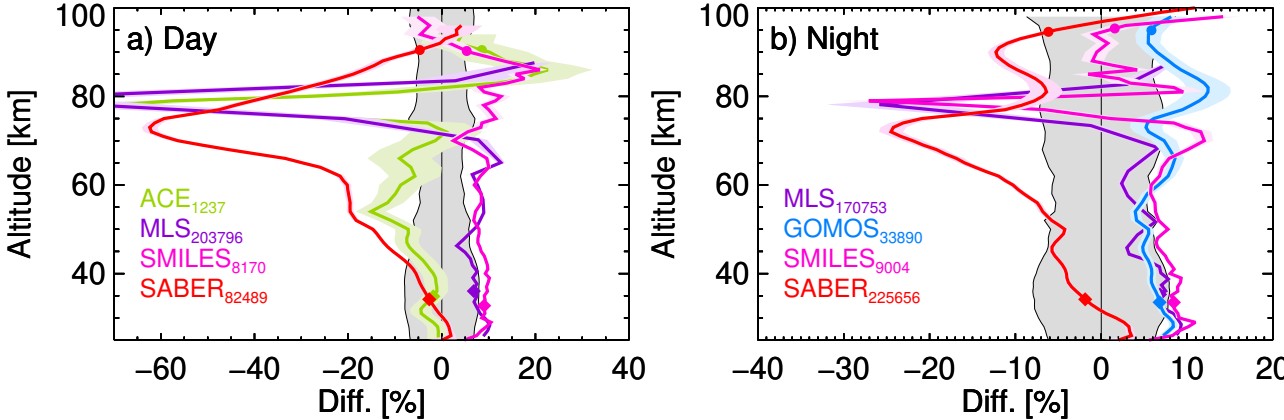

**Figure 12.** Global mean (for all latitudes and seasons) of the daytime $O_3$ vmr differences (MIPAS–instrument) in % of MIPAS between co-located pairs of measurements of MIPAS (MA mode) with ACE-FTS (green), MLS (purple), SMILES (magenta), SABER (red) and GOMOS (light blue). For more details see the caption of Fig. 10.

### 7.2 Results of the comparison

Figures 10 and 11 show the mean daytime and nighttime differences, respectively, between MIPAS and the different instruments (MIPAS−instrument) for the four seasons, grouped in four latitude bins. Further, the global means for all latitudes and seasons but separately for daytime and nighttime are plotted in Fig. 12. We have shown the systematic errors of MIPAS just as a reference.

We see that MIPAS compares very well with ACE in the stratosphere and lower mesosphere (up to 75 km) with differences smaller than 5% except for a narrow region close to 55 km where ACE is larger by about 10%.

The comparison with MLS and SMILES in this region is very consistent, being MIPAS in general between 5% and 10% larger than both instruments up to near 75 km. Note that the differences with respect to MLS at nighttime in this region are slightly smaller, with values close to 5% from 45 km to 75 km. The better agreement between infrared (MIPAS and ACE) with respect to the microwave instruments (MLS and SMILES) might suggest that the differences are caused by inconsistencies between the spectroscopic data in those spectral regions.





MIPAS agrees well with SABER in the lower stratosphere, up to $\sim$35 km. However, above this altitude, at 40 to 60 km, SABER $O_3$ vmr is larger than MIPAS by 20% during daytime and by 10% at nighttime. The differences are larger for the cases

of MLS, SMILES and GOMOS. Rong et al. (2009) found a positive bias in SABER stratospheric $O_3$, version 1.07, which might explain those differences. Also, SABER retrieves $O_3$ from the bands in the 10 $\mu$m region while MIPAS uses up to 50 km the $v_2$ band. We have discussed above that when using the MWs of MIPAS AB band (10 $\mu$m) we retrieve larger $O_3$ concentrations. Thus, this might partially explain the MIPAS and SABER differences in this region. Above 60 km the differences between MIPAS and SABER increase, reaching differences at 60–85 km within 20–60% at daytime, and in the range of 10–25% at

nighttime, MIPAS ozone mixing ratios being always smaller. Smith et al. (2013) already reported that daytime SABER $O_3$ is overestimated in this region. One possible reason for this overestimation is the larger collisional rate (about a factor of two) for the $O_3(v_1, v_3) + M(N_2, O_2)$ process used in SABER with respecto to MIPAS retrievals (Mlynczak et al., 2013; López-Puertas et al., 2018). This would also explain the larger differences at daytime as non-LTE effects are larger for these conditions (Funke et al., 2012).

The large relative differences of MIPAS with respect to ACE and MLS near 80 km are caused by the smaller values of $O_3$; the difference in absolute values is smaller than 0.1 ppmv. MIPAS, however, shows slightly larger ($\sim$10%) $O_3$ vmr than SMILES in this region. In the daytime upper mesosphere, near 85–90 km, it seems that MIPAS has a positive bias of 10 to 20% with respect to ACE, MLS and SMILES, although it is smaller the SABER $O_3$ in similar values. However, the agreement between all instruments near the secondary maximum is remarkably good.

The differences between the instruments for nighttime conditions in the stratosphere and lower mesosphere (Fig. 12b) are consistent with daytime. In these conditions we also included GOMOS, which seems to corroborate the differences found with MLS and SMILES.

Near 60–70 km, the agreement of MIPAS with MLS is better than for daytime and the differences between MIPAS with MLS, GOMOS and SMILES are very similar, and are in the range of 5–10%. The larger differences and of opposite sign with

SABER have been discussed above.

At the altitude of the secondary maximum, the agreement between the three instruments for nighttime conditions, is very good, with differences comprised within $\pm$10%. MIPAS and SMILES $O_3$ are very similar, being GOMOS about 10% smaller and SABER about 10% larger. The secondary peak is practically at the same altitude, $\sim$95 km.

Overall, MIPAS $O_3$ is within 5% and 8% below 70 km with all these instruments except SABER. In this region MIPAS

seems to have a positive bias of 5-8% with respect to MLS, SMILES and GOMOS. The agreement with ACE is better and the differences are within 5% except for a narrow region near 55 km, where ACE is larger by about 10%. SABER seems to have a positive bias in the order of 20-60%, more pronounced during daytime. In the upper mesosphere/lower thermosphere, in the $O_3$ secondary maximum, the agreement between those instruments is, generally, within $\pm$(6–10)%. Near 80 km, around the $O_3$ minimum, the nighttime relative differences are larger, within $-20$ to $+10$%, and are even larger at daytime but very small

in absolute terms (within 0.1 ppmv). In general, the differences between all instruments are practically within the MIPAS $1\sigma$ systematic errors, except for SABER in the mesosphere, and MLS and ACE near 80 km. The latter case however is caused by the very low $O_3$ concentration which explodes the relative differences.





Compared to the difference with these instruments in the previous version V5R_O3_m22 (López-Puertas et al., 2018), we find that the differences with MLS, SMILES and GOMOS are slightly larger (2–3%) in the stratosphere. This is a consequence

of the effort of keeping the MIPAS NOM and MA/UA data as much consistent as possible, in particular of using the same spectroscopic database and microwindows. Opposite, the agreement with SABER in this region has been improved. The comparison with MLS at 60–70 km has improved, probably because of the new version of MLS whose retrieval has been extended up to ~90 km. The agreement with ACE near the stratopause has also been improved (probably also caused by the new version of ACE). Further, the agreement of the new version of MIPAS with all instruments has been significantly improved in the upper

mesosphere/lower thermosphere, reducing the differences from $\sim \pm 20\%$ to $\sim \pm 10\%$. In this respect, the new $O_3$ a priori of MIPAS (particularly in the daytime) and the smaller relaxation/quenching of $O_3(v_1,v_3)$ by atomic oxygen are the reasons for the better agreement. In addition, the diurnal variation of $O_3$ in the upper mesosphere (near 80 km, see Fig. D3) has also been ameliorated.

## 8 Summary and Conclusions

In this work we present the most recent version of the MIPAS middle and upper atmosphere ozone data (versions V8R_O3_561, V8R_O3_661 and V8R_O3_761), covering from 20 up to 100/105 km, retrieved from MIPAS observations in the three middle atmosphere modes (MA, UA and NLC, respectively). This version includes the most recent version 8 level-1b MIPAS spectra, which were processed with a retrieval algorithm that incorporates several improvements over the previous data version (V5R_O3_m22). Among them are: a more accurate retrieved temperature, the treatment of the background continuum and

radiance offset correction, and the selection of optimized numerical settings. Microwindows and spectroscopic data were also update in order to be consitent with the $O_3$ retrieval of the nominal (NOM) mode (Kiefer et al., 2023). Specific to the middle/upper atmospheric $O_3$ are three important updates: the $O_3$ a priori data (particularly important for daytime), the revision of the non-LTE processes, and the atomic oxygen concentration during nighttime.

Another different aspect of this version is the novel treatment of errors, following the TUNER (Towards Unified Error

Reporting) recommendations (von Clarmann et al., 2022), where the different components (random or systematic) of a given error source were propagated independently and also allowing the error propagation of uncertainties in parameters obtained in preceding retrievals (e.g. temperature). The random component of the $O_3$ error is dominated by the spectral noise. Typical values ($1\sigma$ for single profiles) for daytime are smaller than 5% below ~60 km, 5–10% between 60 and 70 km, 10–20% at 70–90 km and about 30% at 95 km. For nighttime, they are very similar than for daytime below 70 km but significantly smaller

above, with values of 10–20% at 75–95 km, 20–30% at 95–100 km and larger than 30% above 100 km. The random $O_3$ error is generally larger than the total systematic uncertainty above around ~60 km.

The systematic error is dominated by the uncertainties of spectroscopic data below ~60 km (~6%), with a significant contribution of the ILS and, to a lesser extent, the persistent part of the gain calibration uncertainty. Above ~60 km the systematic errors are dominated by the non-LTE uncertainties but the spectroscopic data error also contributes significantly under certain

conditions. The non-LTE errors are smaller than 1% below 60 km, 2–8% at 60–85 km, and 5–12% above 85 km. The ILS





contribution is also appreciable at above ∼90 km. As a consequence of the new TUNER method, which takes compensation effects of errors that act via multiple pathways into account, the estimates of systematic errors in the 80-100 km range are significantly smaller than those of version V5R_O3_m22 (see Fig. 5 and Table 4 in López-Puertas et al., 2018). On one hand, non-LTE errors are smaller, because of a proper propagation of the atomic oxygen abundance, but also the $O_3$ errors incurred
by the uncertainties in temperature and in the systematic component of the gain are smaller, as they partially compensate.

When comparing with the previous V5R_O3_m22 version, we note: 1) that $O_3$ abundance is larger in about 2–5% (0.2–0.5 ppmv) at 20–50 km; 2) a decrease by ∼2–4% between 50 and 60 km, mainly in the tropics and mid-latitudes; 3) a more pronounced nighttime $O_3$ minimum just below 80 km leading to a more realistic diurnal variation in this region; 4) a larger (∼40%, 0.2–0.3 ppmv) $O_3$ concentration in the secondary maximum during daytime in the tropical and mid-latitudes; and 5) a
10–30% decrease in the $O_3$ abundance in the secondary maximum at nighttime.

We found that the $O_3$ fields retrieved from the nominal mode (NOM) and the middle atmosphere modes in their common altitude range (20–70 km) are fully consistent, with differences generally being within 1–2%. Only in the lower mesosphere (60–70 km) during daytime $O_3$ NOM seems to be larger than $O_3$ MA/UA in 2–10% for most latitudes.

The comparison performed with the most recent data versions of SABER, GOMOS, MLS, SMILES and ACE-FTS, shows
that MIPAS $O_3$ is within 5% and 8% below 70 km with all instruments except SABER. In this region, MIPAS seems to have a positive bias of 5-8% with respect to MLS, SMILES and GOMOS. The agreement with ACE is better and the differences are within 5% except for a narrow region near 55 km, where ACE is larger by about 10%. SABER seems to have a positive bias in the order of 20–60%, more pronounced during daytime.

In the upper mesosphere/lower thermosphere, the agreement between those instruments is, generally, within ±6–10%. Near
the $O_3$ minimum around 80 km, the nighttime relative differences are larger (within −20 to +10%), and are larger at daytime, but very small in absolute terms (within 0.1 ppmv).

In general, the differences between all instruments are practically within the $1\sigma$ systematic errors of MIPAS, except for SABER in the mesosphere, and MLS and ACE near 80 km. The latter, however, is a result of the very low $O_3$ abundance.

Comparing those differences with those reported by (López-Puertas et al., 2018) for the previous version V5R_O3_m22
(López-Puertas et al., 2018), we find that the differences with MLS, SMILES and GOMOS are slightly larger (2–3%) in the stratosphere. This is the result of using a different spectroscopic database in this version. Conversely, the agreement with SABER in this region has been improved. The comparison with MLS at 60–70 km has improved, probably because the new version of MLS has been extended up to ∼90 km. The agreement with ACE near the stratopause has also improved (likely originated by the new version of ACE). Noticeably, the new version of MIPAS agrees much better than before with all instru-
ments in the upper mesosphere/lower thermosphere, reducing the differences from ∼ ±20% to ∼ ±10%. Further, the diurnal variation of $O_3$ in the upper mesosphere (near 80 km) has ben significantly improved.

*Data availability.* The MIPAS data can be obtained from the KITopen repository. The data will also be available on demand in HARMOZ format (Sofieva et al., 2013, https://essd.copernicus.org/articles/5/349/2013/).



The supplement related to this article is available online at: https://doi.org/10.5194/amt-0-1-2023-supplement.

*Author contributions.* MLP performed the data analysis, wrote the manuscript and had the final editorial responsibility for this paper. MGC performed the comparison of MIPAS data with all the other instruments. BF developed the retrieval setup and performed many test calculations. TvC ensured the TUNER compliance of error estimates. NG was responsible for spectroscopy issues, developed parts of the error estimation software, and carried out some retrieval tests. UG provided and maintained the retrieval software and developed parts of the error estimation software. SK and ALi ran the retrievals. MK coordinated and performed related test calculations and error estimations. ALa con-

tributed to quality control and prepare the data in HARMOZ format. GPS organized the interfacing between IMK and IAA, the consistency between NOM and MA/UA/NLC data, and took care of the quality control. All the authors participated in the development of the retrieval setup, contributed to the discussions, and provided text and comments.

*Competing interests.* At least one of the (co-)authors is a member of the editorial board of Atmospheric Measurement Techniques. The peer-review process was guided by an independent editor, and the authors also have no other competing interests to declare.

*Acknowledgements.* The IAA team acknowledges financial support from the Agencia Estatal de Investigación, MCIN/AEI/10.13039/501100011033, through grants PID2019-110689RB-I00 and CEX2021-001131-S. The Karlsruhe Institute of Technology (KIT) team was supported by the Deutsches Zentrum für Luft- und Raumfahrt (DLR) under contract no. 50EE1547. The computations were partly done in the framework of a Bundesprojekt (grant MIPAS_V7) on the Cray XC40 "Hazel Hen" of the High-Performance Computing Center Stuttgart (HLRS) of the University of Stuttgart.



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



## Appendix A: Microwindows

Table A1 lists the microwindows and altitude ranges used in the retrieval of MIPAS ozone V8R_O3_m61.

**Table A1.** Microwindows and altitude ranges used in the retrieval of MIPAS ozone V8R_O3_m61.

| No. | Wavenumber (cm$^{-1}$) Minimum | Wavenumber (cm$^{-1}$) Maximum | Altitudes (km) |
|-----|---------|---------|----------------|
| 1 | 687.6875 | 688.6875 | |
| 2 | 689.3125 | 691.8750 | |
| 3 | 692.2500 | 695.1875 | |
| 4 | 707.1250 | 710.0625 | |
| 5 | 712.3125 | 713.4375 | |
| 6 | 713.5000 | 716.4375 | |
| 7 | 716.5000 | 719.4375 | |
| 9 | 728.5000 | 729.3750 | |
| 10 | 730.0625 | 730.5000 | |
| 11 | 731.9375 | 732.8750 | |
| 12 | 734.0000 | 734.7500 | |
| 13 | 736.4375 | 739.3750 | |
| 14 | 739.4375 | 741.9375 | |
| 15 | 745.2500 | 745.6875 | |
| 16 | 746.6875 | 747.1250 | |
| 17 | 747.6250 | 748.3750 | |
| 18 | 749.5625 | 752.5000 | |
| 19 | 752.9375 | 755.8750 | |
| 20 | 758.3750 | 759.4375 | |
| 21 | 759.5000 | 761.8750 | |
| 22 | 765.0000 | 765.6250 | |
| 23 | 767.5000 | 768.0000 | |
| 24 | 771.8750 | 772.1250 | |
| 25 | 774.2500 | 774.5625 | |
| 26 | 776.5000 | 776.7500 | |
| 27 | 780.2500 | 781.9375 | |
| 28 | 788.9375 | 789.6875 | |
| 29 | 790.7500 | 791.0000 | |
| 30 | 791.1875 | 791.5625 | |
| 31 | 1034.1250 | 1034.3750 | |
| 32 | 1034.4375 | 1035.0000 | |
| 33 | 1038.1875 | 1039.0000 | |
| 34 | 1040.0000 | 1040.8125 | |
| 35 | 1048.8125 | 1049.5000 | |
| 36 | 1050.6250 | 1051.8125 | |
| 37 | 1053.3125 | 1053.7500 | |
| 38 | 1054.6875 | 1055.4375 | |

(Altitude columns: 18, 21, 24, 27, 30, 33, 36, 39, 42, 45, 48, 51, 54, 57, 60, 63, 66, 69, 72, 75, 78–102)

## Appendix B: Tables of the O$_3$ error budget

In this section we include tables of the O$_3$ error budget for the atmospheric conditions at selected altitudes discussed in Secs. 4.2.1 and 4.2.2.





**Table B1.** Ozone error budget for Tropics day, MA. All uncertainties are $1\sigma$.

| altitude (km) | mean $O_3$ vmr (ppmv) | NLTE (%) | interf (%) | ILS (%) | offset (%) | gain (%) | spectro (%) | T+LOS (%) | noise (%) | random (%) | syst (%) |
|---|---|---|---|---|---|---|---|---|---|---|---|
| 20 | 1.0 | <0.1 | 2.8 | 1.8 | 1.1 | 0.4 | 9.1 | 1.9 | 3.7 | 5.3 | 9.2 |
| 30 | 11 | <0.1 | 0.2 | 2.3 | 0.3 | 1.8 | 6.8 | 0.7 | 1.0 | 1.4 | 7.4 |
| 40 | 6.7 | <0.1 | 0.2 | 2.0 | 0.3 | 1.6 | 6.9 | 0.7 | 1.3 | 1.7 | 7.3 |
| 50 | 2.6 | <0.1 | 0.1 | 2.4 | 0.4 | 0.5 | 5.3 | 0.6 | 2.2 | 2.4 | 5.8 |
| 60 | 1.1 | 0.2 | <0.1 | 1.8 | 1.6 | 0.8 | 5.6 | 2.2 | 6.3 | 7.0 | 5.8 |
| 70 | 0.2 | 5.1 | <0.1 | 0.4 | 1.3 | 0.9 | 6.4 | 2.5 | 12 | 12 | 8.1 |
| 80 | 0.2 | 6.1 | <0.1 | 0.6 | 2.3 | 1.2 | 3.4 | 0.1 | 16 | 16 | 6.9 |
| 90 | 1.0 | 6.7 | <0.1 | 2.2 | 4.8 | 1.0 | 3.3 | 1.3 | 19 | 20 | 7.0 |
| 96 | 0.7 | 11 | <0.1 | 3.6 | 9.3 | 0.9 | 3.9 | 1.7 | 33 | 36 | 9.0 |
| 100 | 0.7 | 10 | <0.1 | 3.5 | 10 | 0.7 | 3.4 | 1.6 | 37 | 38 | 10 |

**Table B2.** Ozone error budget for Tropics night, MA. All uncertainties are $1\sigma$.

| altitude (km) | mean $O_3$ vmr (ppmv) | NLTE (%) | interf (%) | ILS (%) | offset (%) | gain (%) | spectro (%) | T+LOS (%) | noise (%) | random (%) | syst (%) |
|---|---|---|---|---|---|---|---|---|---|---|---|
| 20 | 1.0 | <0.1 | 2.5 | 1.7 | 1.0 | 0.4 | 8.7 | 1.8 | 3.6 | 5.0 | 8.8 |
| 30 | 11 | <0.1 | 0.2 | 2.2 | 0.3 | 1.8 | 6.7 | 0.7 | 0.9 | 1.3 | 7.3 |
| 40 | 6.7 | <0.1 | 0.2 | 2.0 | 0.3 | 1.6 | 7.0 | 0.7 | 1.4 | 1.8 | 7.4 |
| 50 | 2.8 | <0.1 | 0.1 | 2.4 | 0.4 | 0.9 | 5.0 | 0.6 | 2.4 | 2.6 | 5.5 |
| 60 | 1.5 | 0.2 | <0.1 | 1.6 | 1.6 | 1.0 | 4.3 | 2.2 | 5.3 | 6.0 | 4.6 |
| 70 | 1.1 | 5.2 | <0.1 | 1.6 | 2.4 | 1.3 | 4.8 | 5.0 | 9.3 | 11 | 7.0 |
| 80 | 0.5 | 5.9 | <0.1 | 0.6 | 1.3 | 2.1 | 3.6 | <0.1 | 13 | 14 | 6.8 |
| 90 | 8.9 | 7.5 | <0.1 | 0.9 | 2.6 | 1.6 | 2.8 | 2.1 | 11 | 12 | 7.9 |
| 96 | 14 | 11 | <0.1 | 0.8 | 2.8 | 1.1 | 2.6 | 2.7 | 12 | 13 | 11 |
| 100 | 11 | 10 | <0.1 | 1.9 | 4.3 | 0.7 | 2.8 | 2.2 | 17 | 18 | 9.3 |

**Table B3.** Ozone error budget for Northern midlatitude spring day, MA. All uncertainties are $1\sigma$.

| altitude (km) | mean $O_3$ vmr (ppmv) | NLTE (%) | interf (%) | ILS (%) | offset (%) | gain (%) | spectro (%) | T+LOS (%) | noise (%) | random (%) | syst (%) |
|---|---|---|---|---|---|---|---|---|---|---|---|
| 20 | 2.9 | <0.1 | 2.7 | 0.8 | 1.0 | 1.9 | 4.0 | 1.4 | 1.9 | 4.1 | 4.1 |
| 30 | 8.1 | <0.1 | 0.2 | 2.4 | 0.3 | 1.7 | 6.7 | 0.6 | 1.1 | 1.7 | 7.3 |
| 40 | 6.8 | <0.1 | 0.2 | 2.1 | 0.3 | 1.6 | 7.1 | 0.7 | 1.4 | 1.8 | 7.6 |
| 50 | 2.5 | <0.1 | 0.1 | 2.4 | 0.4 | 0.5 | 5.4 | 0.6 | 2.3 | 2.6 | 5.9 |
| 60 | 1.0 | 0.2 | <0.1 | 2.0 | 1.7 | 1.1 | 5.4 | 2.2 | 6.1 | 6.8 | 5.7 |
| 70 | 0.2 | 4.1 | <0.1 | 0.5 | 1.6 | 1.1 | 5.3 | 2.3 | 11 | 11 | 6.6 |
| 80 | 0.2 | 5.9 | <0.1 | 0.7 | 2.2 | 0.9 | 3.6 | 0.2 | 15 | 15 | 6.8 |
| 90 | 0.7 | 5.6 | <0.1 | 2.7 | 6.4 | 1.0 | 3.3 | 1.5 | 23 | 24 | 6.3 |
| 96 | 0.3 | 6.1 | <0.1 | 3.9 | 10 | 0.7 | 3.2 | 1.9 | 36 | 37 | 7.3 |





**Table B4.** Ozone error budget for Northern midlatitude spring night, MA. All uncertainties are $1\sigma$.

| altitude (km) | mean O$_3$ vmr (ppmv) | NLTE (%) | interf (%) | ILS (%) | offset (%) | gain (%) | spectro (%) | T+LOS (%) | noise (%) | random (%) | syst (%) |
|---|---|---|---|---|---|---|---|---|---|---|---|
| 20 | 2.9 | <0.1 | 2.7 | 0.8 | 1.0 | 1.9 | 4.1 | 1.4 | 1.9 | 4.2 | 4.2 |
| 30 | 7.7 | <0.1 | 0.2 | 2.4 | 0.3 | 1.7 | 6.9 | 0.7 | 1.1 | 1.6 | 7.5 |
| 40 | 7.0 | <0.1 | 0.2 | 2.3 | 0.3 | 1.5 | 7.4 | 0.8 | 1.5 | 1.9 | 7.8 |
| 50 | 2.7 | <0.1 | 0.1 | 2.4 | 0.5 | 1.0 | 5.1 | 0.6 | 2.5 | 2.7 | 5.7 |
| 60 | 1.4 | 0.4 | <0.1 | 1.7 | 1.6 | 1.4 | 3.9 | 2.2 | 4.8 | 5.6 | 4.4 |
| 70 | 1.1 | 2.3 | <0.1 | 1.6 | 2.8 | 1.6 | 4.2 | 4.1 | 7.8 | 10 | 4.9 |
| 80 | 0.2 | 4.6 | <0.1 | 0.8 | 1.3 | 0.9 | 3.2 | <0.1 | 14 | 15 | 5.3 |
| 90 | 7.2 | 4.4 | <0.1 | 1.0 | 2.2 | 1.6 | 2.9 | 3.4 | 11 | 12 | 5.1 |
| 96 | 12 | 6.9 | <0.1 | 1.4 | 3.0 | 1.2 | 2.7 | 3.8 | 12 | 14 | 6.9 |
| 100 | 7.3 | 6.7 | <0.1 | 3.1 | 5.4 | 1.2 | 2.8 | 2.7 | 21 | 22 | 7.1 |

**Table B5.** Ozone error budget for Northern polar summer day, MA. All uncertainties are $1\sigma$.

| altitude (km) | mean O$_3$ vmr (ppmv) | NLTE (%) | interf (%) | ILS (%) | offset (%) | gain (%) | spectro (%) | T+LOS (%) | noise (%) | random (%) | syst (%) |
|---|---|---|---|---|---|---|---|---|---|---|---|
| 20 | 2.3 | <0.1 | 3.9 | 1.5 | 1.0 | 2.4 | 4.0 | 1.4 | 2.2 | 5.0 | 4.8 |
| 30 | 4.6 | <0.1 | 0.2 | 2.4 | 0.3 | 1.8 | 5.9 | 0.4 | 1.0 | 1.5 | 6.5 |
| 40 | 5.9 | <0.1 | <0.1 | 1.7 | 0.3 | 1.7 | 5.8 | 0.6 | 1.4 | 1.6 | 6.3 |
| 50 | 2.4 | <0.1 | 0.2 | 2.5 | 0.5 | 1.6 | 4.5 | 0.6 | 2.3 | 2.5 | 5.4 |
| 60 | 1.1 | 0.2 | <0.1 | 1.3 | 1.2 | 1.5 | 4.6 | 1.7 | 3.3 | 4.1 | 4.9 |
| 70 | 0.4 | 4.1 | <0.1 | 1.6 | 2.3 | 0.9 | 6.0 | 2.1 | 8.6 | 9.3 | 7.3 |
| 80 | 0.1 | 7.5 | <0.1 | 0.8 | 2.4 | 0.7 | 4.0 | 1.4 | 15 | 16 | 7.9 |
| 90 | 1.0 | 8.1 | <0.1 | 1.2 | 4.9 | 1.2 | 3.3 | 4.7 | 17 | 19 | 6.6 |
| 96 | 0.6 | 12 | <0.1 | 2.6 | 8.4 | 1.8 | 4.4 | 5.1 | 31 | 33 | 10 |

**Table B6.** Ozone error budget for Southern polar winter night, MA. All uncertainties are $1\sigma$.

| altitude (km) | mean O$_3$ vmr (ppmv) | NLTE (%) | interf (%) | ILS (%) | offset (%) | gain (%) | spectro (%) | T+LOS (%) | noise (%) | random (%) | syst (%) |
|---|---|---|---|---|---|---|---|---|---|---|---|
| 20 | 2.5 | <0.1 | 2.4 | 1.9 | 0.9 | 2.0 | 4.6 | 1.7 | 1.9 | 4.5 | 4.7 |
| 30 | 5.3 | <0.1 | 0.2 | 2.7 | 0.4 | 1.5 | 6.9 | 0.8 | 1.3 | 1.9 | 7.5 |
| 40 | 4.4 | <0.1 | <0.1 | 3.1 | 0.3 | 0.9 | 6.5 | 0.7 | 1.6 | 2.7 | 7.0 |
| 50 | 1.8 | 0.2 | 0.2 | 2.2 | 0.8 | 1.9 | 4.8 | 0.8 | 3.5 | 4.0 | 5.4 |
| 60 | 1.3 | 0.8 | <0.1 | 1.9 | 1.4 | 2.4 | 3.6 | 1.8 | 5.3 | 6.0 | 4.5 |
| 70 | 1.3 | 4.7 | <0.1 | 1.1 | 2.3 | 1.6 | 3.7 | 6.0 | 8.8 | 11 | 5.4 |
| 80 | 0.7 | 4.8 | <0.1 | 0.9 | 1.2 | 2.4 | 4.0 | 1.2 | 12 | 12 | 5.1 |
| 90 | 7.1 | 6.0 | <0.1 | 2.0 | 1.9 | 1.3 | 2.9 | 1.0 | 11 | 12 | 6.5 |
| 96 | 6.2 | 5.5 | <0.1 | 3.7 | 3.6 | 1.2 | 2.7 | 1.8 | 16 | 16 | 6.4 |
| 100 | 3.6 | 5.4 | <0.1 | 5.3 | 6.7 | 1.2 | 2.7 | 2.2 | 27 | 28 | 7.2 |





**Appendix C: Random and systematic components of the O$_3$ errors**

We show in this section the random and systematic components of the different error sources shown in Fig. 5 and discussed in Secs. 4.2.1 and 4.2.2.



**Figure C1.** The random components of the O₃ error budget for MA data shown in Fig. 5 for different atmospheric conditions (top to bottom) tropics, northern mid-latitudes in spring and polar regions, for daytime (left column) and nighttime (right column). All error estimates are 1-σ uncertainties and given in percentage. Error contributions are labelled "T+LOS" for the propagated error from the T+LOS retrieval, "noise" for error due to measurement noise, "spectro" for the spectroscopic error, "gain" for gain calibration error (see text), "offset" for error due to spectral offset (see text), "ILS" for instrument line shape error (see text), "interf" for the uncertainty in the abundance of the interfering species, and "NLTE" for non-LTE related errors. The total random ("random") and systematic ("syst") errors are also shown.

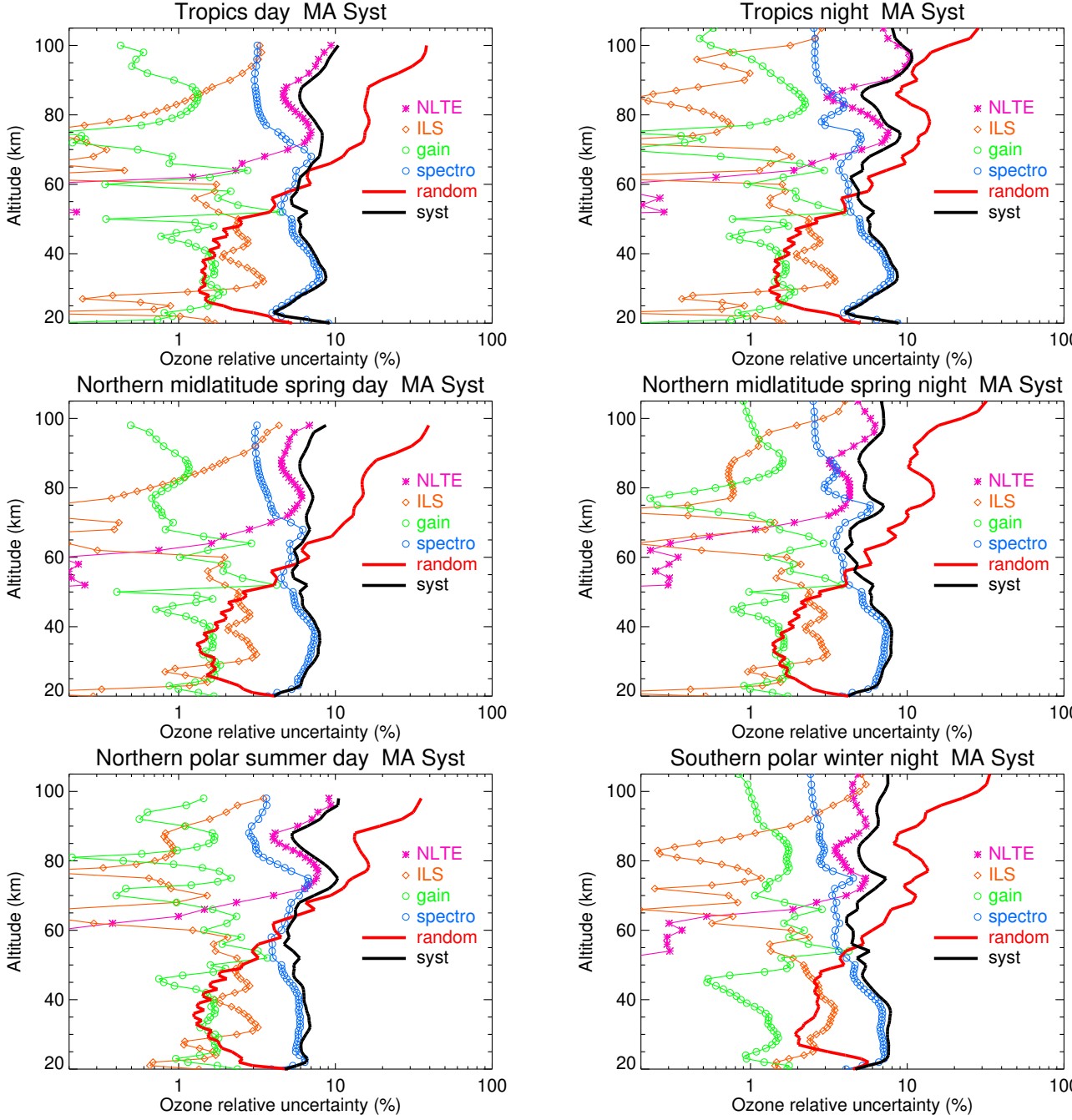

**Figure C2.** As Fig. C1 but for the systematic components of the O$_3$ errors. The total random ("random") and systematic ("syst") errors are also shown.





## Appendix D:  $O_3$  Climatology


We present in this Appendix several distributions of $O_3$, represented as function of altitude, latitude and time, for the new data version, V8R_O3_m61. Similar figures, and the corresponding discussion of the major features in the $O_3$ distributions, were presented by López-Puertas et al. (2018) for the previous version, V5R_O3_m22 of MIPAS $O_3$. The features shown in these figures are very similar. Only the absolute values have changed slightly as have been discussed above in Sec. 5. One significant

improvement, however, is the diurnal variation (see Fig. D3). We refer the reader to López-Puertas et al. (2018) for a detailed description of the $O_3$ fields but recommend to use and cite this new version of MIPAS MA/UA data.

### D1    Monthly zonal mean distributions

Figures D1 and D2 show composite monthly zonal means of MIPAS $O_3$ data for the 2007-2012 period for day- and night-time, respectively. The figures show the characteristic primary, secondary and tertiary maxima and their seasonal evolution (see more

details in López-Puertas et al., 2018).

### D2    Diurnal variation

For completeness, we also show the composite seasonal zonal mean of the MIPAS $O_3$ diurnal differences (10 am–10 pm in percentage of 10 pm) in Fig. D3. The differences near 80 km have been considerably improved in the data version. See general features in López-Puertas et al. (2018).

### D3    Annual variability

Figure D4 shows the annual variability as latitude× months cross sections of $O_3$ at different altitudes for daytime (left column) and nighttime (right column). See López-Puertas et al. (2018) for a description of the major features.

### D4    Altitude-resolved time series

In this section we show the inter-annual variability of altitude-resolved time series of $O_3$ at the tropical and polar latitudes

(Fig. D5); and as latitude×time cross sections at given altitudes (Fig. D6).



**Figure D1.** Composite monthly zonal mean of MIPAS data taken in the MA mode for the 2007-2012 period for daytime (local time of 10 am). White areas denote regions where MIPAS has no sensitivity to measure the very low ozone values. Contours are 0.1, 0.5, 1, 1.5, 2, 4, 6, 8, 10 and 12 ppmv.



**Figure D2.** Composite monthly zonal mean of MIPAS data for the 2017-2012 period for nighttime. Contours are 0.5, 1, 1.5, 2, 4, 6, 8, 10, and 15 ppmv.



**Figure D3.** Composite seasonal zonal mean of $O_3$ diurnal differences (10 am–10 pm in percentage of 10 pm) of MIPAS data taken in the MA mode for the 2007-2012 period. DJF stands for December, January and February; MAM for March, April and May; JJA for June, July and August; and SON for September, October and November. Contours are $-80$, $-50$, $-30$, $-20$, $-10$, $-5$, $-2$, 2, 5, 10 and 20%.

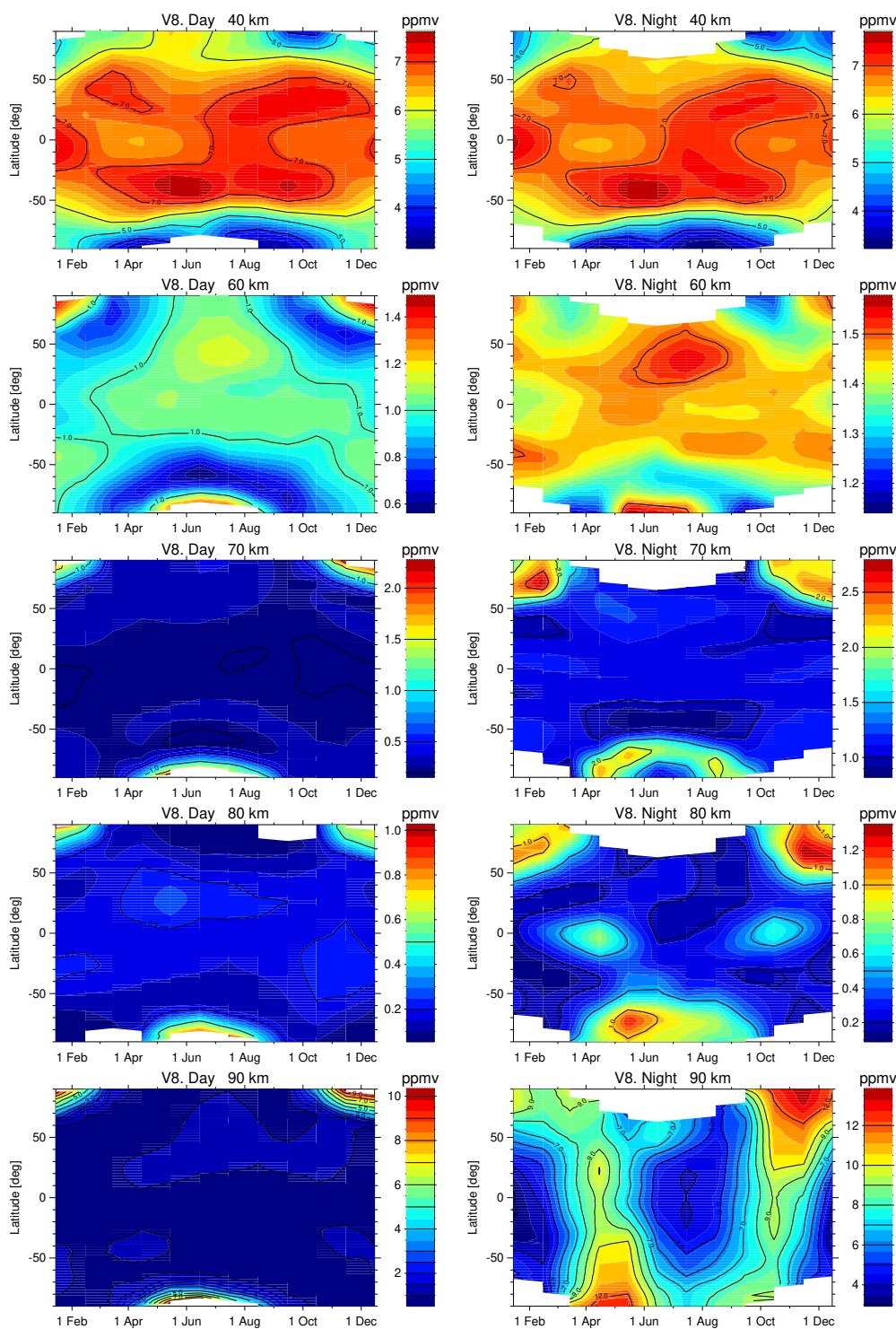

**Figure D4.** Seasonal evolution versus latitude of $O_3$ vmr at different altitudes for daytime (left row) and nighttime (right row). Note the different scales used in the different panels.

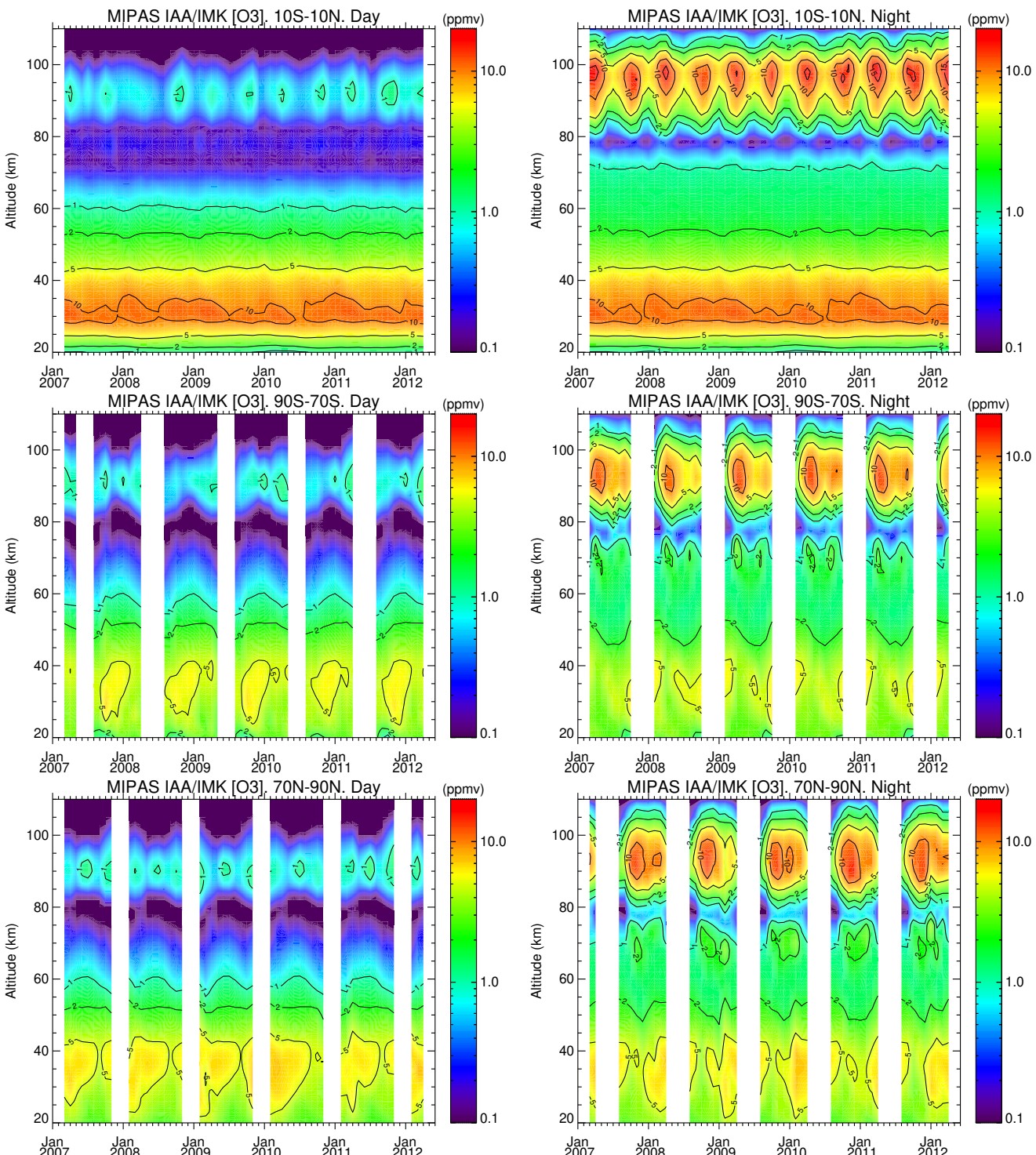

**Figure D5.** Altitude-resolved time series, for latitudes near the equator (10°S-10°N) (top), the Southern polar region, 70°S-90°S, (middle) and the Northern polar region, 70°N-90°N (bottom), for daytime (left column) and nighttime (right column).



**Figure D6.** Cross section of latitude/time MIPAS $O_3$ at 50 km, 70 km and 90 km for daytime (left column) and nighttime (right column). Note the different color scales in some panels.