# Peer review of "MIPAS ozone retrieval version 8: middle atmosphere measurements"

_Atmospheric Measurement Techniques, 2023_

## Author Comment (AC1)

**Response to Chris Boone (Reviewer #1)**

We would like to thank Chris Boone for his very helpful review and constructive comments and suggestions. We have considered all of them and are commenting below some of them. His comments are given below in black and our responses in blue.

This paper seems quite thorough and generally well written, except for a few patches where the English usage could be improved (discussed below). Detailed comments are provided below, none of them particularly concerning.
R. Thank you very much. Please see some responses below. We are also carefully reviewing the English usage, all over and particularly for Sec. 4.2 and 6.

> Line 106: "…along the LOS." The acronym LOS is not defined.
It was defined in a few lines above.

> Line 150: "and the apodization of calculated spectra used a wider frequency range."
What apodization is being used?
R. We use a Norton-Beer strong apodization. This has been added to the text.

> Line 369: "…information on the confidence limits of the error margins…"
I cannot decipher what this means. Are you saying some uncertainties (e.g., for intensities) are unavailable, or you don't know if the errors provided are 1-sigma or 2-sigma, or something else?
R. We meant that it is not clear if the errors provided are 1-sigma, 2-sigma or other. This has been clarified in the revised version.

> Line 333: "The uncertainties of the spectrally interfering molecules with ozone which are not jointly fitted (e.g., as in the case of water vapour), as well as their vertical covariances, are estimated from the error covariance matrices of previous MIPAS data version V5."
I do not see anywhere an indication of which interferers (other than H2O and a passing mention of CO2 laser lines) are being included in the analysis.
R. In addition to CO2 and H2O, the following interfering species were included:
N2O, CH4, NO2, NH3, HNO3, ClO, OCS, HCN, CH3Cl, H2O2, C2H2, C2H6, COF2, C2H4, F-22, CCl4, CFC-113, CFC-114, N2O5, HCFC-141B, HCFC-142B, ClONO2, CH3CCl3, CH3OH, Acetone and PAN. A sentence has been added to the text.

> Line 384: "…ingoing uncertainties affecting the retrieval were reported…"
Perhaps "…various contributions to retrieval uncertainties were reported …"?
Correct. It has been changed.

Starting in the last few paragraphs, the level of English seems worse than the text up to that point. The issue continues for this section (4.2). I will provide no further specific comments on English usage, but I would suggest this section be rewritten to improve grammar and phrasing. The level of English improves again following this section, although Section 6 could also use a mild rewrite for English usage.
R. Thank you very much for the corrections already given. We have carefully reviewed the English usage for the entire manuscript paying particular attention to Secs. 4.2 and 6. We hope it is better readable now.

> Line 396: "T-LOS" Terminology is not used consistently. Sometimes T-LOS, sometimes T+LOS (in the figures), sometimes TLOS. It is not made clear that LOS refers to the line of sight and relates to the tangent altitude determination.
R. Sorry about that. We have revised the whole manuscript and adopted the 'T+LOS' terminology, the same as in the O3 NOM retrieval paper (Kiefer et al., 2023).
LOS is explained in Sec. 4.1.2, the first time we mention it.

> Page 21, caption to Figure 7: Titles on the leftmost panels explicitly assume m = 5, MA measurements (i.e., they indicate 561-522), and the caption assumes m=5 for V8 (V8 561) but leaves m as a variable elsewhere. This creates some confusion. Are you comparing V8 MA results only to MA results from V5 or to results from all other V5 measurement modes? The same question applies to Figure 8.
R. This is a good point, sorry for the confusion. We are comparing only MA data. This has been clarified. Also, as pointed out by the other referee, we have fully revised the versions' nomenclature. In particular all "V8" has been changed to "V8R", as all the middle/upper atmosphere data were taken with the **R**educed spectral resolution. The same applies to "V5", changed to "V5R".

> Figure 7: The color scale for the rightmost panels extends from -10 to +10 %, but there are contours in the plots labelled as 20 and -20. Is the expectation that the reader will mentally extrapolate the color scale (e.g., brighter red means even more positive)?
R. The figures have been redone with extended scales. Also, a note for the still 2 contours going beyond the colour scale in the upper/right panel has been included in the legend.

> Line 566: "...the differences are caused by inconsistencies between the spectroscopic data in those spectral regions."
It has been shown that O3 intensities in HITRAN 2016 (the basis of ACE v4 retrievals) are too weak by about 3% (doi:10.1016/J.JQSRT.2019.01.004), consistent with this assumption.
R. Thank you very much for the reference. A sentence has been included in the revised version stating that although that difference does not fully explain the differences between ACE-FTS and MLS/SMILES, it goes in the right direction.

> Line 666: data availability should probably include all the data plotted in the paper: MLS, ACE-FTS, SMILES, etc. Some links are provided in the text but are more appropriately listed here. ACE-FTS has no data availability information provided. The link provided for MLS is broken.
R. Thank you for checking the links. They have been updated, the ACE-FTS link has been included and all the links are listed now in the "Data availability" section.

All other minor corrections and typos have been included/corrected.

---

## Author Comment (AC2)

**Response to Reviewer #2**

We would like to thank the reviewer for his/her extremely careful reading. His/her comments and suggestions are very helpful and constructive. We have taken all of them into account and are answering his/her questions below. His/her comments are given below in black and our responses are in blue.

The authors appear to have been careful and thorough, and the paper is essentially publishable as it stands. I have just one significant concern which is the ad-hoc adjustment of the k3 chemical quenching reaction rate (L187-189) which seems to have been largely on the basis of bringing the retrieved O3 more into line with other instruments. While this is not unreasonable, I would like this highlighted in the abstract so that future researchers, perhaps also with similarly high O3 measurements, will not automatically assume that they must be incorrect since it disagrees with all previous results.

R. Thank you very much for the very good judgement of our work. We have no problem at all in bringing this up in the abstract. We have included the following sentence in the abstract:
*"In particular, the collisional relaxation of O3(v1,v3) by the atomic oxygen was reduced by a factor of two in order to obtain a better agreement of nighttime mesospheric O3 with ``non-LTE-free'' measurements."*
Nevertheless, let us clarify that there has been a misunderstanding. The rate reduced by a factor of 2 was not k3 (the chemical removal of O3), but the **collisional** relaxation of the vibrationally excited state O3(v1,v3) with atomic oxygen, e.g., O3(v1,v3) + O <-> O3 + O. In order to make this clear we have now written the full collisional relaxation process explicitly (see process k_vt_O in the revised Table 1).

Further, in order to clarify this point, the chemical removal of the vibrationally excited state O3(v1,v3) with O, i.e., O3(v1,v3) + O <-> O2 + O2 should not be confused with reaction 3, i.e., the chemical removal of O3 in the ground state (where most of O3 molecules are) by O.
Note also, as we state in the manuscript, that the chemical removal of the vibrationally excited state O3(v1,v3) with O is negligible and it was already neglected in the previous version V5R (see the manuscript and also López-Puertas et al., 2018). From the non-LTE point of view, e.g., for computing the population of the emitting states O3(v1,v3), there is no practical difference between loss of O3(v1,v3) by collisional deactivation, i.e., O3(v1,v3) + O <-> O3 + O, or by chemical loss, O3(v1,v3) + O <-> O2 + O2. Actually, in laboratory measurements, we cannot discern between those two processes (see West et al., 1978).

Minor comments

1) L32: Rather than 'ample', it would be more informative to give the actual spectral range.
R. Included

2) L51-52: It would be helpful to have a table giving the details of the scan patterns used for these three modes (which could, for example, incorporate additional information such as total number of days of each and horizontal spacing). Also, instead of L81-83.
R. Effectively, the information on the altitude range and steps for the three modes was given a few lines below, in L81-83. The information on the number of measurements and their temporal

distribution is already given. Nevertheless, we have included references to a couple of web pages of Oxford University where all this information is listed in tables.

3) I found the nomenclature for the different versions of L1b and the ozone product inconsistent and confusing. Eg L74 implies that 'V8' is being used to refer to L1B v8.03 and in the next line it is also referred to as 'V8R'. Similarly does V5 refer to L1B or O3?
 L79 then refers to O3 'version 7' but it is unclear which L1B data is used.
R. The referee is fully right. Unless one knows all MIPAS details it was confusing. As the referee well guessed, the first part of the version name refers to the version of level 1b spectra, the first number of the second part of the name (after '_') refers to the observation mode (as explained in the footnote on page 2), and the last two numbers to the L2 version.
The first part is composed of the level 1b version, 5, 7 or 8, and a letter `H' or `R´ referring to the spectral resolution or the phase of the mission, H for high resolution, phase 1, or R for Reduced resolution, phase 2.
The confusion with the O3 version name referring to the level 1b version was caused because all of the middle atmosphere measurements MA, UA and NLC were taken in phase 2 (**R**educed resolution, VxR), and there are no middle atmosphere measurements taken during phase 1, e. g. with **H**igh spectral resolution.
**Action**: To avoid confusion we have changed, all over in the text and figures, the level 1b part of the O3 version name, from V5, V7 or V8 to  V5R, V7R or V8R.

It is not helped by varying references to 'L1b', 'level 1b' and 'level-1b'  and 'level 1' data. L92-93 - again unclear whether V8R, V5R refer to L1b or ozone products. Some figure captions (7,8) also revert to long  notation.
R. Again, the referee is right. We have harmonized all these different names to 'level 1b'.
Figure captions (7,8)  have been consistently revised.

4) Is a '2-points horizontal temperature gradient' the same as a 'linear  temperature gradient' ?
R. Yes, it is. For clarity, "2-points" has been changed by "linear" in the revised version.

5) L111-112: this is confusing. Do you mean that the offset is assumed to be same for all microwindows in each band, with just altitude variation?
This seems a strange assumption given that the offset is presumably of unknown origin and therefore probably has some spectral dependence.  Anyway, please rephrase more clearly.
R. Your interpretation is correct but we wrongly wrote that the offset is frequency-independent. Both the continuum and offset are frequency and altitude-dependent. This has been corrected and re-written in the revised version.

Also, there should be some definition within this paper (eg Table A1) of the spectral ranges of the A and AB bands.
R. The definition of the spectral ranges of the A and AB bands were (are) a few lines below. Nevertheless, they have also been added in Table A1.

6) The spectral structure that the forward model is required to resolve is the Doppler broadened O3 line. It would be useful to have this mentioned at this point, along with a value for this.

R. As stated in the discussion paper, we tested the now-used forward model grid of 0.00097656 cm-1 grid (very close to 0.001 cm-1) against the previous finer grid of 0.0005 cm-1 and found negligible differences in the retrieved O3 (see more details below)
Action: We have included a sentence in the revised version stating that the 0.00097656 cm-1 grid is sufficient to model the O3 Doppler-broadened lines.

I didn't understand what you meant by the grid for absorption cross-sections. - I assumed this would be the forward model grid, but if not, that suggests some further interpolation is required (and the figure 0.00097656... seems oddly specific - is there a reason for this number?)
R. The text was not correct and confusing. We are using in V8R a grid in the forward model of 0.00097656 cm-1. This figure was chosen, for convenience, as the number which is closer to 0.001 cm-1 and fulfils that the ratio of the spectral resolution before numerical apodization, 0.0625 cm-1 to this figure is equal to $2^N$, being N an integer. In our case N=6, or 0.0625/0.00097656=64=$2^6$.
This paragraph has been rewritten.

By my calculation the 0.001cm-1 grid resolution just matches the O3 Doppler half-width, so I'm slightly surprised this is adequate unless your grid is adapted to the line-centres rather than a fixed grid.
R. See the tests discussed above about this point that confirm its adequacy.
After hearing about your concern we were equally concerned and have performed some further simple tests. First, we checked your calculation of the O3 Doppler half-width and we could confirm that it was correct. Then, we tested the sensitivity of the retrieval with respect to the gridwidth and the position of the gridpoints relative to the line center. We found that, regardless if the line's center coincides with a grid point or is placed anywhere between two grid points, the ratio of the areas of the monochromatic Doppler line and that of the line sampled at 0.001 cm-1 is smaller than 1e-5. It is noticeable however that if doubling this grid, e.g., 0.002cm-1, then the differences rise very quickly to 2-5%, depending on the position of the line's centre wrt the grip points. (Figures are available on request).
Hence, with these tests, we can confirm that a spectral grid of 0.001 cm-1 is adequate, with maximum errors in the order of 1e-5.

7) I may be wrong, but if your Tikhonov regularisation is applied to the shape of the VMR profile then it seems it would change the total ozone amount retrieved due to the non-linearity between VMR and partial column amount, even if the regularisation maintains the same average local VMR value.
R. The referee is right, although this implies a higher-order effect. in order to be precise, we have rewritten the sentence changing the "O3 total amount" to "the retrieved O3 (vmr) profile". Changed sentence: *"We recall that the Tikhonov regularization chosen does not systematically push the retrieved ozone profile towards the a priori but only constrains the shape of the vertical profile."*

8) Table 1: it would be clearer to put the actual rate constants in this table, along with assumed uncertainties used in the error budget.
R. Done, see revised Table 1. Apparently, the referee is confounding the chemical reactions affecting the concentration of O3 with the processes controlling the population of the excited emitting states O3(v1,v3). Actually, one of them is common, the O3 production, O+O2+M, where

O3 is produced with a given nascent distribution into the O3(v1,v2,v3) states. For this reason, we have extended the table to include the two other major processes that drive the non-LTE population of O3(v1,v3), the collisional deactivation with N2 and O2 (M), process 5 in Table 1, and with atomic oxygen, O, process 6 in Table 1.

9) Eq. 1 shows the k2 reaction being neglected but the text seems to refer to this reaction simply as 'chemical production', which seems an obscure way of referring to it.
R. Apologies. That was a mistake. It now reads: *"... where the chemical loss of O3 by H has been neglected."*.

10) L406. Does this mean by two 'identical' instruments? which includes calibration errors, forward model assumptions etc. In that case errors due to interfering species would also be the same, so not contribute to the random error.
R. No, we do not mean two 'identical' instruments. Calibration, forward model assumptions, etc. would cause mainly a bias, not a standard deviation of the differences. Errors due to interfering species are not the same, because the two instruments may measure in different spectral regions and might use other information on the interfering species.

Typography/Grammar
R. Thank you very much for the meticulous reading! We have corrected/included all your suggestions, except one: the months for solstice and equinox.

General comment: frequent use of hyphen (-) rather than en-dash (-- in LaTeX) to indicate a range of numbers.
R. We have that in mind when writing but a few of them escape our attention. Thanks!

L178: Did you really mean 'e.g.' here? Or should it be 'i.e.'? In other words, are there species other than O and H?
R. Very subtle. Thanks! Corrected.

Fig 1: 'solstice' would better describe NDJ rather than DJF, similarly 'equinox'.
R. You are probably right but the majority of the works in the literature refer to DJF and MAM. We have not changed the monthly distribution.